



# Turbulent heat flux dynamics along the Dotson and Getz ice-shelf fronts (Amundsen Sea, Antarctica)

Blandine Jacob[1], Bastien Y. Queste[1], and Marcel D. du Plessis[1]

[1]Department of Marine Sciences, University of Gothenburg, Gothenburg, Sweden

**Correspondence:** Blandine Jacob (blandi.jacob@gmail.com)

**Abstract.**

In coastal polynyas, where sea–ice formation occurs, it is crucial to have accurate estimates of heat fluxes in order to predict future rates of sea–ice formation. The Amundsen Sea Polynya is the fourth largest coastal polynya around Antarctica, yet remains poorly observed because of its remoteness. Consequently, we rely on models and reanalysis that are unvalidated to

study the effect of atmospheric forcing on polynya dynamics. We use summer ship-board data from the NBP22/02 cruise to understand the turbulent heat flux dynamics in the Amundsen Sea Polynya and evaluate our ability to represent these dynamics in ERA5. We show that cold and dry air outbreaks from Antarctica enhance air–sea temperature and humidity gradients, triggering episodic heat loss events. The heat loss is larger along the ice shelves, and it is also where the ERA5 turbulent heat flux exhibits the largest biases, underestimating the flux by up to 141 W m$^{-2}$ due to its coarse resolution and misrepresentation

of ice-shelf location. By reconstructing a turbulent heat flux product from ERA5 variables using a nearest neighbour approach to obtain sea surface temperature, we decrease the bias to 107 W m$^{-2}$. Using a 1D-model, we show that the mean co-located ERA5 heat loss underestimation of -28 W m$^{-2}$ led to an overestimation of the summer evolution of sea surface temperature (heat content) by +0.76 °C (+8.2×10$^{7}$ J) over 35-days. By obtaining the reconstructed flux, the reduced heat loss bias (12 W m$^{-2}$) reduced the seasonal bias in sea surface temperature (heat content) to -0.17 °C (-3.30×10$^{7}$ J) over the 35-days. This

study shows that caution should be applied when retrieving ERA5 turbulent flux along the ice shelves, and that a reconstructed flux using ERA5 variables shows better accuracy.

## 1 Introduction

Among other properties such as momentum, gas and moisture, the atmosphere and the ocean exchange heat, which maintains the Earth's energy balance (Yu, 2019). The climate is highly controlled by the ocean notably because the ocean has the ability

to absorb heat from the atmosphere and to redistribute it poleward (Bigg et al., 2003). The ocean is thus the largest heat sink on Earth, absorbing 91 % of the excess heat due to greenhouse gases (Forster et al., 2021). The exchange of heat between the ocean and the atmosphere – or air–sea heat flux – is therefore a crucial process to predict the current and future weather (through heat and moisture released in the atmosphere), upper-ocean physics (sea surface temperature SST variability, sea–ice formation, heat content HC in the mixed-layer), climate (e.g. teleconnection such as el Niño) and the ensuing impacts on society (e.g.

agriculture, health, water resource) (Cronin et al., 2019). Because of air–sea heat fluxes' importance, the scientific community





calls for reducing uncertainties to have better flux estimates (Cronin et al., 2019). By increasing the number of observations, our understanding of fluxes can be enhanced and the associated uncertainties reduced (Cronin et al., 2019; Swart et al., 2019; Yu, 2019; Bourassa et al., 2013).

Polar regions are poorly observed because of their remoteness and harsh conditions, which implies we have a particular lack
of understanding of the flux dynamics there. In particular, the Southern Ocean south of 60° S has been identified by Swart et al. (2019) as a targeted observation region for the ongoing decade. The Amundsen Sea, West Antarctica, is a shelf sea south of 60° S seasonally covered by sea–ice and with few historical observations. However, scientific efforts have been concentrated there recently, for example through the International Thwaites Glacier Collaboration (Turner et al., 2017; Scambos et al., 2017), due to adjacent melting glaciers and one of the most biologically productive coastal polynyas (the Amundsen Sea Polynya;
ASP) in the Antarctic (Arrigo and van Dijken, 2003). In comparison to the surrounding sea–ice that acts as a lid, the polynyas operate as open windows that enable direct exchange with the atmosphere (Smith Jr and Barber, 2007). Consequently, the polynyas usually inherit the "sea–ice factory" nickname due to the sea–ice formation caused by heat loss. To be able to better predict sea–ice formation in the Amundsen Sea, we therefore need to improve our knowledge of heat exchange in the ASP.

Without air–sea heat flux observations available, previous studies in the Amundsen Sea or other Antarctic coastal seas have
relied on global reanalyses products (e.g., Kumar et al., 2021; Zhou et al., 2022; Yu et al., 2023; Papritz et al., 2015). A global climate reanalysis product combines observations and past forecasts through data assimilation, providing gridded data with a regular temporal resolution. ERA5 reanalysis, produced by the European Centre for Medium–range Weather Forecasts (ECMWF; Hersbach et al., 2020) and its predecessor (ERA–Interim) are considered the most robust reanalyses in Antarctica (Bromwich et al., 2011; Bracegirdle and Marshall, 2012) and in the Amundsen Sea (Jones et al., 2016; Jones, 2018). However,
ERA5's ability to reproduce the flux magnitude and variability in the Amundsen Sea, particularly near important boundaries such as ice shelf fronts, has not been validated.

The air–sea heat flux has two components: the radiative flux and the turbulent flux. We hypothesize that the Amundsen Sea has a high potential for turbulent loss due to cold dry air and relatively warm SST in summer (above the freezing temperature). In this study, we perform the first study of turbulent heat flux (THF) in the Amundsen Sea based on in situ observations and
we (i) identify the temporal and spatial variability of the THF from shipboard observations, (ii) assess ERA5 accuracy at representing these fluxes, and (iii) investigate the relative importance of THF on the summer evolution of SST.

## 2 Data and methods

### 2.1 Data

#### 2.1.1 Observations: shipboard and glider data

The meteorology and underway system of the research vessel recorded variables listed in Table 1 during 57 days. We use these observations to compute the THF; the computation method is described in Sec. 2.2.1. The ship departed Punta Arena (Chile) on 6 January 2022, reaching the Amundsen Sea (72° S, 117° W) on 15 January 2022. It then entered the polynya region and



spent 31 days within 20 km of Dotson or Getz ice shelves, and finally left the polynya region on 25 February 2022 (Fig. 1b).
To determine the THF and be consistent with ERA5's temporal resolution, we compute hourly averages of the variables in
Table 1. Hourly position of the ship is used to create a classification: Southern Ocean, open ocean in the polynya region (ship
more than 20 km away from the coastline), in front of Dotson or Getz ice shelves (ship within 20 km), and in a sea–ice covered
region (where the sea–ice concentration; SIC; is larger than 0.15 %). Airflow distortion caused by the superstructure of the
RV *Nathaniel B. Palmer* on the wind speed values was negligible (Appendix A, Fig. A1), we therefore did not perform any
correction.

| Variable | Unit | Sensor | Height [m] |
|---|---|---|---|
| Air temperature | °C | R.M. Young 41372LC | 19.2 |
| SST | °C | Sea–Bird SBE 38 | $\sim$ -5 |
| Wind speed | m s$^{-1}$ | Gill 1390-PK-062 | 34.4 |
| Relative humidity | % | R.M. Young 41372LC | 19.2 |
| Longwave radiation | W m$^{-2}$ | Eppley PIR | 33.78 |
| Shortwave radiation | W m$^{-2}$ | Eppley PSP | 33.78 |

**Table 1.** Sensors installed and variables used in this study recorded by the RV *Nathaniel B. Palmer*.

We use Conservative Temperature, Absolute Salinity and pressure from an ocean profiling Seaglider that was deployed
during the ship campaign to compute the daily HC in the upper 40 m of the water column. The glider was deployed on 17
January 2022 in front of Dotson ice shelf (73.8° S, 112.6° W). The glider then headed south towards the ice shelf, and returned
north along the Dotson–Getz trough, before being recovered on 4 February 2022 (red transect, Fig. 1a). A total of 286 profiles
were collected. The data are gridded horizontally per profile and vertically with a resolution of two meters.



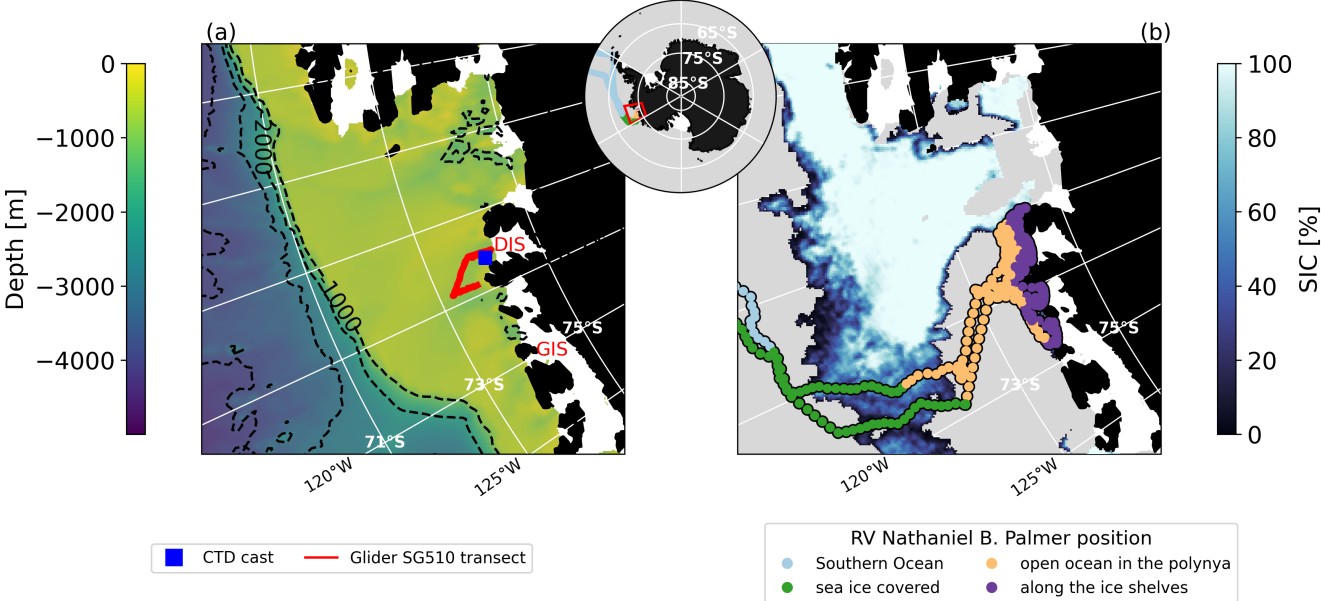

**Figure 1.** (a, b): Antarctica in black, ice shelves from BedMachine (Morlighem et al., 2020) in white. (a): Background is bathymetry from RTopo (Schaffer et al., 2016), the blue square is the CTD cast location used to force the 1D model, in red is the glider transect. Dotson and Getz ice shelves (DIS; GIS) are indicated in red. (b): sea–ice concentration from ASI algorithm on 19 February 2022 (downloaded from https://data.seaice.uni-bremen.de; Spreen et al. 2008), the RV transect is indicated by the scatters colored by the location classification. The red box on the zoomed-out map is the Amundsen Sea plotted on (a, b).

### 2.1.2 Reanalysis dataset: ERA5

In this study, we assess ERA5 by comparing its hourly mean THF with the THF computed from the observations. We also use some of ERA5's hourly mean meteorological and sea surface variables to recalculate the THF (Table 2). ERA5 is a global reanalysis product that provides "maps without gap" of atmospheric and sea surface variables (Hersbach et al., 2020). It has a hourly temporal resolution and a regular 0.25° lat-lon grid. To co-locate its variables to the ship data, we use the nearest neighbour grid cell and the corresponding hour (as the ship data have been hourly averaged).



| Variable | Unit |
|---|---|
| 2 meter temperature | K |
| SST | K |
| 2 meter dewpoint temperature | K |
| Wind (zonal and meridional components) | m s$^{-1}$ |
| Mean surface net shortwave radiation flux | W m$^{-2}$ |
| Mean surface net longtwave radiation flux | W m$^{-2}$ |
| Mean surface latent heat flux | W m$^{-2}$ |
| Mean surface sensible heat flux | W m$^{-2}$ |

**Table 2.** ERA5 variables used in this study

### 2.1.3 Satellite based data: sea–ice concentration from the ARTIST sea–ice algorithm

We use sea–ice concentration to determine when the research vessel was surrounded by sea–ice (15 % threshold) for the location classification (Fig. 1b). We select the satellite based sea–ice product ARTIST sea–ice (ASI) created by Bremen University (Spreen et al., 2008, data downloaded from: https://data.seaice.uni-bremen.de/amsr2). The ASI algorithm takes input data from the Advanced Microwave Scanning Radiometer 2 (ASMR2) (Level 1B) - a sensor operating on the JAXA satellite GCOM-W1 - and outputs gridded data (Level 3, grid space is 3.125 km or 6.25 km). The temporal resolution is daily, the selected output grid space for this study is 3.125 km.

### 2.2 Methods

### 2.2.1 Turbulent heat flux computation and analyses: COARE 3.5 algorithm, Reynolds decomposition, and indices of contribution

The THF is the sum of the latent heat flux (LHF) and the sensible heat flux (SHF). The LHF is related to the air–sea heat exchange originating from the evaporation whereas the SHF arises from the air–sea temperature gradient. We use the Coupled Ocean–Atmosphere Response Experiment (COARE) 3.5 algorithm (Edson et al., 2013) through AirSeaFluxCode (Biri et al., 2023) to compute the THF from the observations. The COARE 3.5 algorithm relies on bulk parameterisations: the LHF and SHF are computed as a function of air density ($\rho_{air}$), 10-meters wind speed ($U_{10}$), transfer coefficient ($C_q$ or $C_t$) and, for the SHF (Eq. (1)), specific heat capacity ($C_p$), air–sea temperature gradient ($T_{air} - T_{skin}$); and for the LHF (Eq. (2)), latent heat of vaporization ($L_v$), air–sea humidity gradient ($q_{air} - q_{sat}$).

$$SHF = \rho_{air} C_p C_t U_{10}(T_{air} - T_{skin}) \tag{1}$$



$$\text{LHF} = \rho_{air} L_v C_q U_{10}(q_{air} - q_{sat}) \tag{2}$$

The AirSeaFluxCode applies a logarithmic profile to adjust the wind from the measured height to 10 meters. The COARE 3.5 algorithm requires wind speed relative to the ocean surface. Here, we assume that the ocean currents are low in comparison to the wind speed and neglect them. The flux convention is downward, which means a negative (positive) flux corresponds to a heat loss (gain) for the ocean.

We perform a Reynolds decomposition to analyze the flux variability. We decompose SHF and LHF into the sum of their

average (denoted by $^-$) and their anomaly (denoted by '): SHF = $\overline{\text{SHF}}$ + SHF', and LHF = $\overline{\text{LHF}}$ + LHF'. We follow the same method as in Tanimoto et al. (2003); Chuda et al. (2008); Yang et al. (2016) except that we do not neglect the contribution of $C_q'$ and $C_t'$ as they are more than 5 % of their mean values (not shown; criteria following Cayan, 1992). We replace the variables $U_{10}$, $\Delta T = T_{air} - T_{skin}$, $\Delta q = q_{air} - q_{sat}$, $C_q$ and $C_t$ in Eq. (1) and (2) by the sum of their mean and anomaly (details in Appendix B). Finally, we obtain:

$$\text{SHF'} = \text{SHF} - \overline{\text{SHF}} =$$

$$\rho_{air} C_p \left[ \underbrace{\Delta T' \bar{U} \bar{C}_t}_{\text{t-term}} + \underbrace{\overline{\Delta T} U' \bar{C}_t}_{\text{u-term}} + \underbrace{\overline{\Delta T} \bar{U} C_t'}_{\text{ct-term}} + \underbrace{\bar{C}_t (\Delta T' U' - \overline{\Delta T' U'})}_{\text{t-u-term}} \right.$$

$$\left. + \underbrace{\bar{U}(\Delta T' C_t' - \overline{\Delta T' C_t'})}_{\text{t-ct-term}} + \underbrace{\overline{\Delta T}(U' C_t' - \overline{U' C_t'})}_{\text{u-ct-term}} + \underbrace{\Delta T' U' C_t' - \overline{\Delta T' U' C_t'}}_{\text{cov-term}} \right] \tag{3}$$

$$\text{LHF'} = \text{LHF} - \overline{\text{LHF}} =$$

$$\rho_{air} L_v \left[ \underbrace{\Delta q' \bar{U} \bar{C}_q}_{\text{q-term}} + \underbrace{\overline{\Delta q} U' \bar{C}_q}_{\text{u-term}} + \underbrace{\overline{\Delta q} \bar{U} C_q'}_{\text{cq-term}} + \underbrace{\bar{C}_q (\Delta q' U' - \overline{\Delta q' U'})}_{\text{q-u-term}} \right.$$

$$\left. + \underbrace{\bar{U}(\Delta q' C_q' - \overline{\Delta q' C_q'})}_{\text{q-cq-term}} + \underbrace{\overline{\Delta q}(U' C_q' - \overline{U' C_q'})}_{\text{u-cq-term}} + \underbrace{\Delta q' U' C_q' - \overline{\Delta q' U' C_q'}}_{\text{cov-term}} \right] \tag{4}$$

LHF' and SHF' are the flux variability around the mean. We establish indices of contribution for each term of Eq. (3) and (4), following Yang et al. (2016). To quantify the contribution of a term $X$ ($X$ has to be substituted by one of the terms defined in Eq. (3) and (4)) to the flux anomaly $Y$ ($Y$=L for the LHF' /=S for the SHF'), we compute the absolute value of the $X$-term

and divide it by the sum of the absolute values of the seven terms (Eq. (5)).

$$C_Y(X) = \frac{|\text{X-term}|}{\sum |\text{all terms}|} \tag{5}$$

Therefore the contribution indices $C_Y(X)$ have values between 0 and 1 and their sum equals to 1. The closer to 1 is $C_Y(X)$, the larger the contribution of the term $X$ is to the flux anomaly SHF' or LHF'.



### 2.2.2 Turbulent heat flux impact on the sea surface temperature and the heat content: 1D-model

The 1D mixed-layer model PWP (Price Weller Pinkel; Price et al., 1986) is used to investigate the relative impact of different THF estimates, using observations and ERA5, on the SST and HC. The model needs two input files: one contains the initial ocean state (temperature and salinity profiles), the other contains a time series of the atmospheric forcing (radiative flux, turbulent flux, freshwater flux and momentum flux). The initial ocean profile (Fig. C1) comes from a CTD cast in front of Dotson ice shelf (74.0° S, 113° W), Fig. 1a, blue rectangle. We carry out four simulations with forcings from the observations

and ERA5 that differ only in the THF (Fig. C2e, g) in order to isolate its effect. The four simulations are further detailed in Sec. 3.3. We remove the Southern Ocean data to focus on the polynya region. The simulations are accomplished with the aim of: (i) evaluate if the ERA5 flux co-location/computation method is important, (ii) verify if any bias was induced by a moving ship, (iii) retrieve daily change in SST and HC due to THF and compare them to the observations from the glider and the ship.

## 3 Results

### 3.1 In situ observations: turbulent heat flux characteristics in the Amundsen Sea

### 3.1.1 Turbulent heat flux variability: the leading component in the Amundsen Sea

First, we analyze the THF computed from the ship observations (Fig. 2) to understand the heat flux magnitude and variability in the Amundsen Sea (Fig. 3).







**Figure 2.** Main sea surface and atmospheric variables. The colors on top of the first time series is the classification of the RV *Nathaniel B. Palmer* position. The grey shaded area is the heat loss event selected for the case study (Sect. 3.1.3). The blue area in (f) represents the wind direction when blowing from 135 to 225° i.e. blowing from the south-west, south and south-east.





The Amundsen Sea (comprising the polynya region and along the ice shelves in the classification) lost on average more turbulent heat (-52 $\pm$ 51 W m$^{-2}$) than the Southern Ocean region (-30 $\pm$ 26 W m$^{-2}$, Fig. 3e). The largest instantaneous heat loss events were also observed in the Amundsen Sea (maximum is -230 W m$^{-2}$ versus -145 W m$^{-2}$ in the Southern Ocean). In particular, more than 97 % of the turbulent heat loss events larger than 150 W m$^{-2}$ occurred when the ship was within 20 km's of Dotson or Getz ice shelves (purple classification, Fig. 3).

**Figure 3.** Ship observations of (a, b): latent heat flux ; (c, d): sensible heat flux ; (e, f) the sum of the two latter: the turbulent heat flux. A positive heat flux is a gain by the ocean (downward convention). The colors on top of the first time series and the grey shaded area are as in Fig. 2.

Within the polynya region, the THF variations (Fig. 3e) were linked to short-scale SHF loss between -90 and -140 W m$^{-2}$ (Fig. 3c), while in the Southern Ocean, the largest turbulent heat loss events (Fig. 3e) were driven by the LHF loss (Fig. 3a).

       Throughout the time series, the LHF contributed most to the net THF loss, accounting for an average of 57 % of the THF. This is evidenced by the mode of both the LHF and THF being between -30 and -10 W m$^{-2}$, whilst for the SHF the mode is





between -10 and 10 W m$^{-2}$ (Fig. 3b, d, f). Thus, while the LHF contributed the most to the total THF, the most significant
short-term (days to weeks) THF loss events were imputed to large SHF losses.

### 3.1.2    Turbulent heat flux decomposition: enhanced air–sea temperature and humidity gradients responsible for large episodic heat loss events

Below, we investigate the key drivers in the variability of the THF. We decompose the flux anomalies into different terms (Eq. (3), (4)). These terms indicate the contribution of the temperature gradient (t-term), humidity gradient (q-term), wind speed (u-
term), transfer coefficients (ct-term and cq-term), and the cross-contribution of these variables: second-order terms (t-u-term, q-u-term, u-cq term, etc) and third-order term or residual (covariance-term) to SHF' and LHF' (Fig. 4).





**Figure 4.** (a) mean SHF' and (b) mean LHF' in black rectangles binned by 20 W m$^{-2}$ or 40 W m$^{-2}$ (for the first and the last bins). The numbers on top of the black rectangles are the mean values of the SHF' (a) and LHF' (b) for the corresponding bin. The colored bars inside the black rectangles are the different terms from Eq. (3) and (4). Their sum gives SHF' and LHF'. Note that the x-axis is different for the two subplots because the bin (-130, -90) is empty for LHF'.

The t-term (associated with the anomalous air–sea temperature gradient) was larger than the sum of all the other terms, (Fig. 4a, blue colorbars dominate). The indices of contribution of the terms (Table 3) were calculated for each time step and range between 0 and 1 to show the relative importance of one term compared to another. The t-term was the most frequent dominating factor: 37 % of $C_S(t)$ values were above 0.5 (Table 3). This indicates that in 37 % of the data, air–sea temperature gradients were responsible for more than 50 % of SHF'. Following the same arguments, the q-term (anomalous air–sea humidity gradient)






contributed the most to LHF' (Fig. 4b - blue colorbars), but to a slightly lesser extent: above 33 % of LHF' values had $C_L(q)$ as the strict dominant factor.

| $C_S(X)$ | $C_S(t)$ | $C_S(u)$ | $C_S(ct)$ | $C_S(t-ct)$ | $C_S(t-u)$ | $C_S(u-ct)$ | $C_S(cov)$ |
|---|---|---|---|---|---|---|---|
| Median | 0.43 | 0.17 | 0.06 | 0.09 | 0.16 | 0.02 | 0.04 |
| IQR | 0.29 | 0.26 | 0.04 | 0.09 | 0.10 | 0.03 | 0.02 |
| % values > 0.5 | 37.22 % | 1.75 % | 0 % | 0 % | 0 % | 0 % | 0 % |
| $C_L(X)$ | $C_L(q)$ | $C_L(u)$ | $C_L(cq)$ | $C_L(q-cq)$ | $C_L(q-u)$ | $C_L(u-cq)$ | $C_L(cov)$ |
| Median | 0.40 | 0.22 | 0.08 | 0.07 | 0.12 | 0.03 | 0.03 |
| IQR | 0.28 | 0.19 | 0.08 | 0.07 | 0.09 | 0.04 | 0.02 |
| % values > 0.5 | 33.21 % | 7.36 % | 0 % | 0 % | 0 % | 0 % | 0 % |

**Table 3.** Median and interquartile range (IQR) on the indices of contribution of the seven terms of the Reynolds decomposition. $C_L(X)$ is the contribution of the variable anomaly $X$ to LHF', $C_S(X)$ is the contribution of the variable anomaly $X$ to SHF'. An indice of contribution higher than 0.5 means that the term is the strict dominant factor.

The decomposition indicates that the variability of the air–sea property gradient was the dominant factor impacting the variations of the THF. Further investigations show that the atmospheric variables (air temperature and humidity) control the air–sea property gradients. Indeed, the variability of the air temperature is higher (standard deviation (STD) = 2.62 °C) than the variability of the SST (STD = 1.13 °C), Fig. 2b, c. The same statement holds for the humidity: the air humidity has a higher variability (STD = 0.70 g kg$^{-1}$) than the saturated humidity (STD = 0.32 g kg$^{-1}$), Fig. 2d, e. The difference of STD is even larger when we remove the Southern Ocean data (not shown). These results indicate the importance of cold dry air mass driving large heat loss events.

### 3.1.3 Case study of the heat loss mechanism in the Amundsen Sea: cold and dry southerlies trigger the heat loss

To further analyze the heat loss mechanism in the Amundsen Sea, we investigate the relationship between the turbulent heat loss and the broader scale synoptic variability (Fig. 5).





**Figure 5.** (a–c): THF plotted by the wind direction colored by: (a) the temperature gradient, (b) the humidity gradient or (c) the wind speed. The stars correspond to the heat loss event studied and depicted on map (d). (d) is the Amundsen Sea, the coastline is in white. The black dot is the research vessel position on 19 February 2022. For the same day at 11:00, and from ERA5, are plotted the 10 meters wind speed and direction (arrows), the 2 meters air temperature (background), the isobars (black contour).

The large turbulent heat losses were associated with winds blowing from the south (Fig. 5a, b, c) with large temperature Fig. (5a) and humidity (Fig. 5b) gradients. We focus on one heat loss event that lasted six hours on 19 February 2022 from 08:00 to 13:00 (stars on Fig. 5). The THF remained below -170 W m$^{-2}$ (Fig. 3), reaching its peak of -211 W m$^{-2}$ at 11:00. The mean value over the 6 hours was -186 W m$^{-2}$. The wind was directed from the continent (Fig. 5d wind vectors, and Fig. 2f) and brought cold (on average -9.4 °C, Fig. 2b) and dry (1.4 g kg$^{-1}$, Fig. 2d) air on top of the warmer (on average -0.1 °C, Fig. 2c) and moister (3.7 g kg$^{-1}$, Fig. 2e) sea: triggering the heat loss event. A low pressure center was also visible on the map.





It may have enhanced the heat loss event. This weather system (cold and dry continental winds) has been observed for all the major turbulent loss events occurring during this research cruise (not shown). Contrariwise, the instances when the THF was significantly positive ($> 30$ W m$^{-2}$) are consistent with warm and moist northerlies blowing over the Amundsen Sea (Fig. 3a, b and Fig. 2f).

This indicates the role of large scale atmospheric variability on the local flux events. Next, we review start-of-the-art reanal-
ysis to understand the ability of numerical weather models to represent these key processes.

## 3.2 ERA5 reanalysis: revealing the product bias in the Amundsen Sea

### 3.2.1 Turbulent heat flux bias at land–sea boundaries

The research vessel spent 72 % of its time in the polynya region and along the Dotson and Getz ice shelves where few validations of ERA5 have been conducted. First, we analyze the THF from ERA5 by comparing them to the COARE 3.5 fluxes
from the observations (Fig. 6).

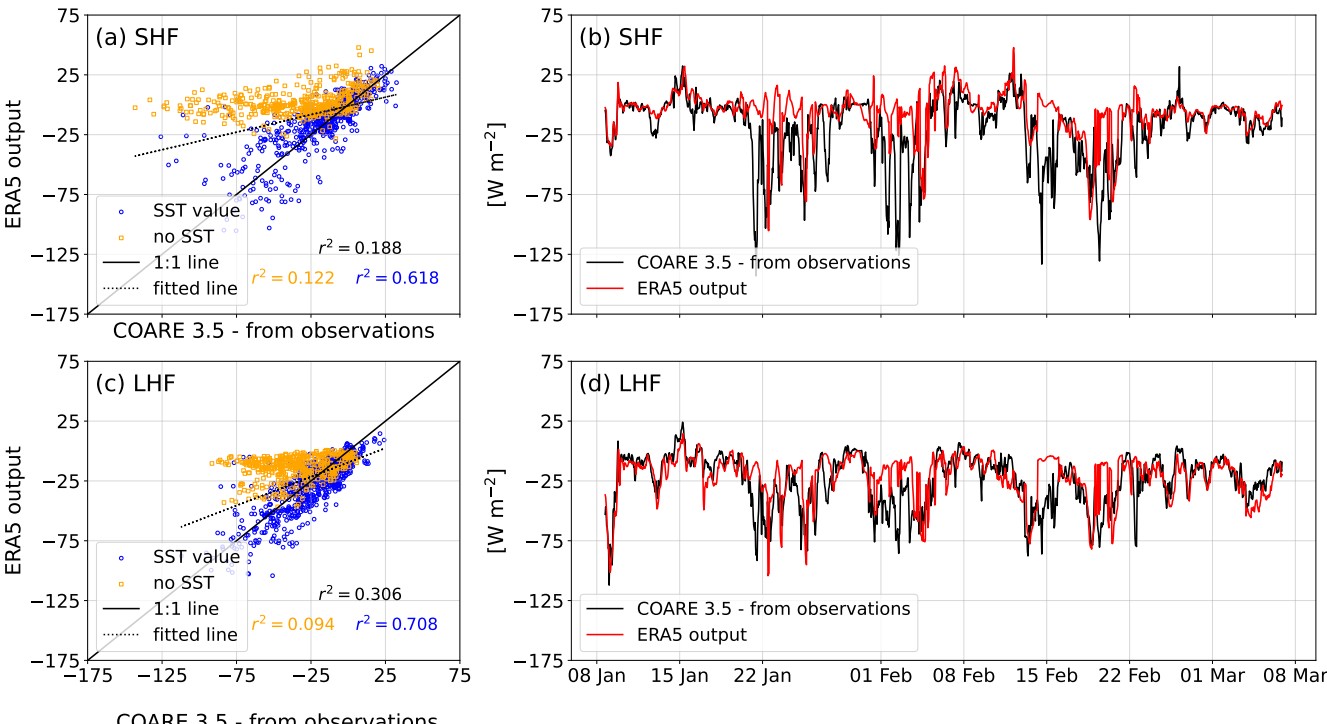

**Figure 6.** (a, b): SHF and (c, d): LHF. (a, c) are colored by a SST mask (in blue the data points where the ERA5 cell has a SST value, in yellow where SST is NaN). The $r^2$ in black is the coefficient of determination for all the scatters, the $r^2$ in blue and in yellow are the coefficients of determination of the scatters corresponding to the SST mask. (b, d): the black lines are the fluxes computed from the research vessel measurements with the COARE 3.5 algorithm, in red are fluxes from ERA5. ERA5 fluxes are co-located to the research vessel position by selecting the nearest ERA5 value.





The agreement between THF product from ERA5 and the THF calculated from in situ data via COARE 3.5 was low; $r^2 = 0.188$ for the SHF (Fig. 6a) and $r^2 = 0.306$ for the LHF (Fig. 6c). Additionally, ERA5 was positively biased in comparison to the observations: the mean SHF and LHF were higher by 13 and 3 W m$^{-2}$ (Table 4, 1st and 2nd rows). Similarly, hourly episodic heat loss events were not well represented by ERA5 (Fig. 6b, d): the difference between the two time series was up to 185   141 W m$^{-2}$ for the SHF and 81 W m$^{-2}$ for the LHF (Table 4, 2nd row).

| Flux product | mean LHF | mean SHF | LHF max diff. | SHF max diff. |
|---|---|---|---|---|
| observation-based | -25 W m$^{-2}$ | -20 W m$^{-2}$ | - | - |
| ERA5 output | -22 W m$^{-2}$ | -7 W m$^{-2}$ | 81 W m$^{-2}$ | 141 W m$^{-2}$ |
| ERA5 from hybrid ds | -29 W m$^{-2}$ | -23 W m$^{-2}$ | 69 W m$^{-2}$ | 107 W m$^{-2}$ |

**Table 4.** Comparison of the different flux products. The LHF and SHF max diff. are the maximum absolute difference between the flux from the observations and another flux product (either the ERA5 fluxes or the ERA5 hybrid fluxes).

The low agreement between the two flux products is explained by a misrepresentation of the location of the ice shelf in ERA5. The research vessel was often stationed along the ice shelves where the ERA5 closest grid cell was considered as ice shelf, and as such does not have a SST value (Fig. 7). The correlation between ERA5 and the COARE 3.5 fluxes improved when only comparing instances where the nearest ERA5 grid cell had a SST value not set to NaN ($r^2 = 0.618$ for the SHF and $r^2 = 0.708$ for the LHF, Fig. 6a, c, blue scatters). For instances where there is no SST, the correlation is weak ($r^2 = 0.122$ 190   for the SHF and $r^2 = 0.094$ for the LHF, same Figure, sub-panels, yellow scatters). Thus, the ERA5 reanalysis THF product underestimates the turbulent losses at the land–sea boundary formed by the ice shelves due to missing SST values. These results illustrate the importance of careful analysis when investigating ice shelf processes in reanalyses.



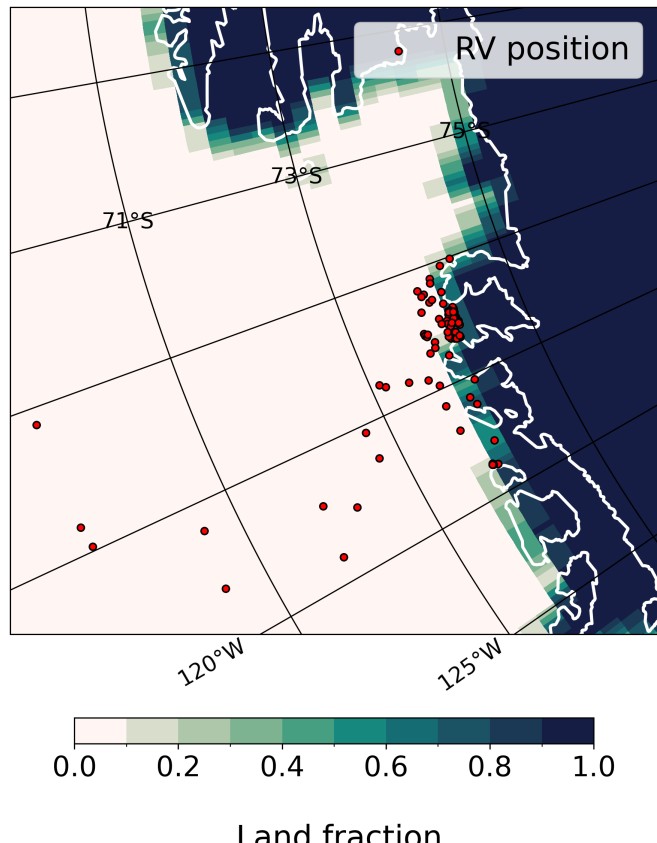

**Figure 7.** Research vessel position plotted every 12 hours (red scatters) and land–sea mask from ERA5 (background). The coastline is in white.

### 3.2.2   A hybrid dataset to reduce the bias

As shown above, the dominant mechanism impacting the THF was the variations in air temperature and humidity. As such, to make use of all available ship based observations to compare with ERA5, we require more suitable method to reduce the SST-based biases identified.

We create an ERA5 dataset with the atmospheric variables (wind, air temperature, dew-point temperature, pressure) co-located using the closest ERA5 value, and with the SST co-located using the closest cell that has a SST value. We found this

reasonable as the SST variability is less important (range = 1.9 °C) than the air temperature (range = 12.2 °C) in the polynya region during the research cruise (Fig. 2b, c). Therefore the gradient of temperature in Eq. (1) mainly depends on the air temperature. The COARE 3.5 algorithm was then applied to compute turbulent fluxes using the new dataset as input (Fig. 8). We refer to the original ERA5 THF product as the "nearest neighbour product", which suffers from an innacurate landmass, and the new product calculated from ERA5 atmospheric variables and valid SST as the "hybrid product" (hybrid because of

the difference of co-location method between the SST and the other variables).



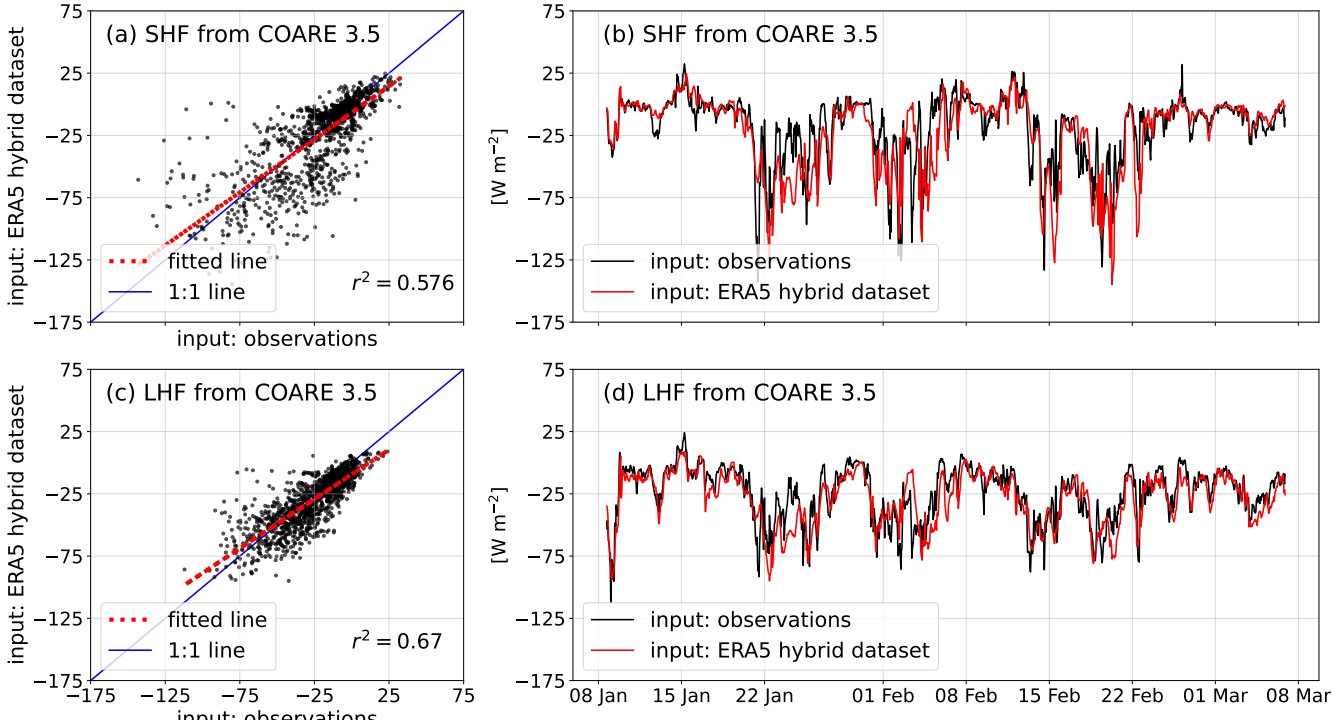

**Figure 8.** (a, b): SHF, (c, d): LHF from COARE 3.5 algorithm with input from ERA5 hybrid dataset (in red) and input from RV *Nathaniel B. Palmer* measurements (in black).

The SHF correlation between ERA5 and the observations was higher with the ERA5 hybrid dataset ($r^2 = 0.576$, Fig. 8a) than the original nearest-neighbour dataset ($r^2 = 0.188$, Fig. 6a). The same statement holds for the LHF: $r^2 = 0.670$ for the hybrid dataset (Fig. 8c) versus $r^2 = 0.306$ for the nearest-neighbour flux (Fig. 6c). The hybrid heat fluxes were also on average closer to the observations-based fluxes but with a negative bias: the mean LHF and SHF for the hybrid-based fluxes were lower by 3 and 4 W m$^{-2}$ than fluxes from the observations (Table 4) because of colder air–temperature (not shown). The instantaneous heat loss was slightly better represented (Fig. 8b, d in comparison to Fig. 6b, d) with a maximum difference between the time series of 107 W m$^{-2}$ for the SHF and 69 W m$^{-2}$ for the LHF (Table 4).

### 3.3   1-D model simulations and glider data: determining the importance of an accurate turbulent heat flux estimate

We understood the characteristics of the heat loss events in the Amundsen Sea (Sect. 3.1) and evaluated ERA5 in this region (Sect. 3.2). We found an underestimation of the turbulent heat loss from ERA5 THF output, and we created a hybrid ERA5 dataset that overestimates the heat loss but fits better the observations. In this last section we evaluate the impact of the THF on the SST and HC variability. More specifically, we determine whether the overestimation (or underestimation) of ERA5 fluxes is critical for estimates of SST and HC. We use a 1-D model to answer these questions. We ran four PWP simulations with different atmospheric forcing that differed only in their THF (Fig. C2e, g and Fig. 9a). Three out of the four THF products were





used earlier in this study: the THF computed from the research vessel via COARE 3.5 ("observation dataset"), the ERA5 THF co-located using the nearest-neighbour approach (suffering from the land–mask issue) and the THF computed from the ERA5 hybrid dataset via COARE 3.5. The last THF is obtained directly from ERA5 at a single ocean grid cell so as to represent the time varying dynamics and exclude any lateral processes associated with a moving ship. We call this the stationary dataset.

### 3.3.1   Turbulent heat flux effect on sea surface temperature

The PWP model predicted a warming of the water column over the 35-days for all four simulations (Fig. 9g), which is in agreement with the seasonal warming expected during the austral summer. However, the SST of the four PWP simulations diverge (Fig. 9c). At the end of the 35-days run (duration of the expedition in the polynya region), the nearest-neighbour SST (darker red line) was higher (0.76 °C, Fig. 9e) than the other three simulations. The hybrid (middle-red line) and stationary (lighter red line) simulations were colder (-0.17 °C and -0.14 °C) than the observation simulation (black line in Fig. 9c). This is

in agreement with the ERA5 overestimation of heat loss for the hybrid dataset and underestimation for the nearest-neighbour dataset. The HC in the 40-meters upper layer was higher by $8.2 \times 10^7$ J at the end of the 35-days for the nearest-neighbour simulation, and lower by $-3.30 \times 10^7$ J and $-2.98 \times 10^7$ J for the hybrid and stationary simulations (Fig. 9f) in comparison to the observation-based simulation.





**Figure 9.** (a) THF input for the PWP model, (b) THF difference for the observation based simulation minus one of three other simulations. (c) SST model output and (d) HC in the upper 40-meters layer computed from the model's output. (e) and (f) are the difference of SST and HC between the observation-based run and the other runs. (g) shows the initial temperature profile (dotted line) and the final temperature profiles. The grey shaded area on all the panels corresponds to a two-days period that we examine in the results.

The observation and the nearest-neighbour simulations had large THF difference (on average 28 W m$^{-2}$ and instantaneously up to 200 W m$^{-2}$ ; Fig. 9b, dark red scatters) that led to an increase in the slope of the difference of SST (Fig. 9e) and HC





(Fig. 9f) between the two simulations. For example, from day 13.5 to day 15.5 (grey shaded area), the nearest-neighbour THF was on average 82 W m$^{-2}$ warmer than the observed THF. This difference explained a SST increase of 0.20 °C and a HC increase of $1.4 \times 10^7$ J (dark red line, Fig. 9e, f) of the nearest-neighbour simulation in comparison to the observation-based simulation.

We note that the daily change in SST ($\Delta SST$, Table 5) has the same order of magnitude between the stationary dataset-based simulation (mean = 0.003 °C d$^{-1}$, STD = 0.042 °C d$^{-1}$) and the hybrid dataset-based simulation (mean = 0.002 °C d$^{-1}$, STD = 0.040 °C d$^{-1}$), which gives us confidence in the credibility of using the PWP model with data that are not stationary (i.e. biases introduced when comparing datasets from moving vessels).

    Thus, the four simulations diverge because of the different THF inputs (indeed all the other atmospheric forcings are identi-
cal). The resulting difference in seasonal evolution of SST (HC) of 0.76 °C ($8.2 \times 10^7$ J) is not negligible for a coastal polynya region where sea–ice formation, ice shelf melting and primary production are dominant processes.

### 3.3.2    A sea surface temperature and heat content bias as large as the spatial-scale variability

In this final subsection, we look at ship-based thermosalinograph data (TSG) (for the SST) and nearby glider data (for the HC) to understand how the temperature change in the PWP simulations compare to the change of temperature in the observations.

$\Delta SST$ from the PWP model output (solid lines, Fig. 10a) and from the TSG observations (dotted line) showed a similar average change (Table 5 ; mean: $O(10^{-3})$ °C d$^{-1}$), except for the nearest-neighbour PWP output that has a higher mean (0.029 °C d$^{-1}$). Regarding the HC, the nearest-neighbour simulation exhibits a higher mean as well, even though the difference is less important than for the SST (Table 6). STD of the daily change of SST and HC are one order of magnitude higher for the observations (0.140 °C d$^{-1}$, $2.0 \times 10^7$ J d$^{-1}$) than for the model outputs ($O(10^{-2})$ °C d$^{-1}$, $O(10^6)$ J d$^{-1}$), Tables 5 and 6.
The larger spread in the observations can be explained by the horizontal processes that are assumed to be non-negligible in a sea-shelf environment, and that were not represented in the 1D model.

    It is thus reasonable to assume that the horizontal processes average out (as seen by the comparison of the means), the seasonal (35-days) evolution of SST and HC in the observations was well represented by the PWP simulations, the nearest-neighbour simulation set aside.





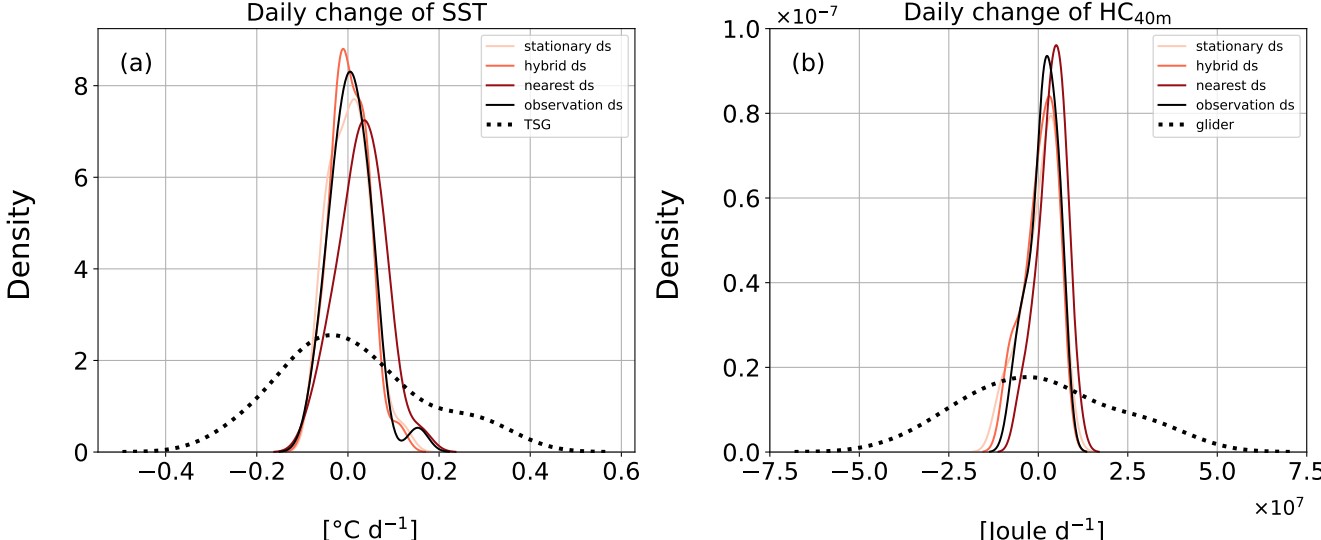

**Figure 10.** Distribution of: (a) the daily change of SST and (b) the daily change of HC in the upper 40-meters layer for the four PWP simulations (continuous line) and the observations (dotted line). For the observations, we used ship-based data at 6 meters for the SST and glider data for the HC.

| Dataset | Mean $\Delta SST$ [°C d$^{-1}$] | STD $\Delta SST$ [°C d$^{-1}$] |
|---|---|---|
| TSG | 0.010 | 0.140 |
| PWP output - obs. | 0.007 | 0.046 |
| PWP output - hybrid | 0.002 | 0.040 |
| PWP output - nearest | 0.029 | 0.049 |
| PWP output - stat. | 0.003 | 0.042 |

**Table 5.** Statistics of the change of temperature $\Delta SST = SST_{t_{i+1}} - SST_{t_i}$ with $t_{i+1} - t_i = 1$ d. STD is the standard deviation. The SST is taken at depth = 6 meters for the PWP model to match the depth of the TSG measurements.



| Dataset | Mean $\Delta$HC [J d$^{-1}$] | STD $\Delta$HC [J d$^{-1}$] |
|---|---|---|
| Glider | $4.9\times10^5$ | $2.0\times10^7$ |
| PWP output - obs. | $1.4\times10^6$ | $3.8\times10^6$ |
| PWP output - hybrid | $5.3\times10^5$ | $4.2\times10^6$ |
| PWP output - nearest | $3.8\times10^6$ | $3.6\times10^6$ |
| PWP output - stat. | $6.1\times10^5$ | $4.7\times10^6$ |

**Table 6.** Same as previous table but statistics of the daily change of heat content in the upper 40-meters layer.

Coming back to the two-days example of the previous section, according to the PWP model we see a misrepresentation of the THF of about 80 W m$^{-2}$ (Fig. 9b) that lead to a SST increase of 0.2 °C (Fig. 9e) and a HC increase of $1.4\times10^7$ J (Fig. 9f). Across this particular heat loss event, these SST and HC increase was of the same order of magnitude as the spatio-temporal variability scale observed in the 35-days ship-based SST observation (STD = 0.140 °C d$^{-1}$, Table 5, Fig. 10a) and 18 days glider data (STD = $2.0\times10^7$ J d$^{-1}$; Table 6, Fig. 10b) that were imputed to horizontal processes. Therefore, the cumulative effect of
small differences in the mean $\Delta$SST and $\Delta$HC due to a bias in the THF is critical for the temporal-scale variability of SST and HC in the Amundsen Sea. 1D processes alone cannot explain the change of SST and HC that we see in the observations. However, our work highlights that a consequent bias in the THF can lead to errors in the estimation of the 35-days evolution of the SST and HC that are of the same order of magnitude of the variability due to horizontal processes.

## 4   Discussion and conclusions

This work provides the first air–sea THF observation-based study in the Amundsen Sea. Large scale cold and dry winds moving from the Antarctic continent over the sea enhance the air–sea temperature and humidity gradients and trigger turbulent heat loss events up to -230 W m$^{-2}$. We show that THF from ERA5 are inaccurate at ice shelf boundaries (mean underestimation of the heat loss of 28 W m$^{-2}$ over the 35-days of the research survey in the polynya region). This can be improved by recalculating fluxes from ERA5 sea surface and atmospheric variables, with vigilance regarding the SST that is sometimes set to NaN due
to ERA5 land–sea mask. The THF obtained is overestimating the heat loss by 12 W m$^{-2}$ on average in the polynya region. As a result of misrepresenting the THF, we show that the seasonal evolution of the modelled SST would be overestimated (+ 0.76 °C at the end of the 35-days PWP simulation) with ERA5 THF, or underestimated (- 0.17 °C) with ERA5 recalculated flux. Therefore, caution has to be made when selecting turbulent fluxes for modelling studies in a coastal polynya where, for example, sea–ice formation and primary production are important processes.





## 4.1 Polynya turbulent flux system

### 4.1.1 The role of the winds

Antarctica is known for its strong surface winds. It might be plausible that the wind contributes significantly to the heat loss by increasing turbulent mixing, yet we did not observe any pattern of wind intensity related to the heat loss (Fig. 4 and Fig. 5c). In our study some caveats regarding the wind processing have to be mentioned. (i) A systematic positive bias in the residuals was observed in Fig. A1 meaning that either ERA5 underestimates the wind speed or the measurements overestimate it. (ii) We considered the wind distortion bias from the superstructure of the ship negligible and decided to not remove any wind data points (Appendix. A), this may have induced a bias. (iii) The currents were neglected in the flux computation. Regarding point (i), we imputed the residual bias to ERA5 as it has been shown that the wind speed along Antarctica coastline is underestimated by ERA5 (Caton Harrison et al., 2022). For (ii), we found that the air–sea property gradients dominate the air–sea THF variability, a lesser importance is placed on the wind speed and the final bias on the THF is assumed to be low. Finally, for point (iii), Kim et al. (2016) have shown from recording current meters installed on a two-years mooring that the coastal surface current in the Amundsen Sea is about 0.2 cm s$^{-1}$, which is 0.03 % of the mean wind speed (7.9 m s$^{-1}$, Fig. 2a) in our dataset. It was therefore reasonable to consider the surface currents negliglible for the THF computation.

### 4.1.2 The role of the broad atmospheric system

The properties of the air (cold and dry) transported by the winds are more important for the THF variability than the wind speed itself. The advection of cold and dry air that triggers ocean heat loss is similarly observed in the Southern Ocean near the Polar Front, at 54° S, 89° W (Ogle et al., 2018). Papritz et al. (2015) showed from ERA-Interim data that the Amundsen Sea is a hotspot for cold air outbreaks (CAOs), which contribute to the turbulent loss. This highlights the importance of having an understanding of the broader atmospheric circulation on the air–sea turbulent heat exchange in the Southern Ocean and Antarctic waters.

As shown, the atmospheric system is thought to be a key component in the Amundsen Sea. Indeed, Jones (2018) used the MetUM model to perform episodic (2 to 3 days) high heat flux study cases in the East Amundsen Sea and found that strong easterlies and southeasterlies associated with cyclones are typically linked to heat loss. The importance of low pressure systems in cold air outbreaks is a result that has also been found in the Ronne polynya (Weddell Sea) from aircraft observations (Fiedler et al., 2010). The Amundsen Sea is known for being a cyclogenesis region and we indeed saw recurring low pressure centers in our study area (Fig. 5d). In particular the Amundsen Sea Low (ASL) is a quasi stationary low pressure center that oscillates between the Ross Sea (to the West) and the Bellingshausen Sea (to the East). Hosking et al. (2013) showed that the ASL influences the meridional component of the large scale atmospheric circulation. When the ASL is positioned to the west, it enhances the southerly flow. The ASL was located at the edge of the Ross and Amundsen Sea (to the west) in January and March 2022 and in front of Thurston Island, in the Amundsen Sea, in February 2022 (the ASL index position was downloaded from https://github.com/scotthosking/amundsen-sea-low-index). The ASL had therefore likely enhanced the southerly (continental) winds over the time span of the research vessel presence in the Amundsen Sea, and consequently increased the turbulent heat



loss through enhanced air–sea temperature and humidity gradients. It would be interesting in future studies to investigate the longer-term air–sea THF variability associated with ASL locations.

### 4.1.3 Spatial variability of the fluxes

In this study, the largest heat losses occurred in front of the ice shelves. This result could indicate that the heat loss is larger along the ice shelves than in the open water in the polynya, but could also be explained by the fact that the research vessel spent 27 days out of 41 in front of the ice shelves when being in the polynya region. However, it has been shown by modelling studies in other coastal polynyas (Renfrew et al. 2002, polynya that forms off the Ronne Ice Shelf (Weddell Sea) and Jones 2018, Pine Island Glacier polynya, Pope Smith Kohler polynya and Thurston polynya (Eastern Amundsen Sea)) that the THF decreases with fetch. When continental cold (dry) air is advected on top of the sea, it gains heat (moisture) from the ocean as it travels offshore, which reduces the gradient of temperature (humidity) and as the consequence the SHF (LHF). This result gives us confidence regarding the spatial variability observed in our dataset and illustrates the key mechanisms for driving heat loss in regions of ice shelf dynamics.

### 4.2 Assessing ECMWF turbulent heat flux in the Amundsen Sea

From ERA-Interim data, Papritz et al. (2015) created a CAO climatology. They found that the CAOs summer frequency is under 3 %, and consequently focused on the nonsummer months. In autumn winter and spring, they found that the CAOs contribute to the turbulent heat loss enhancement. Their study focused on the region of the Amundsen Sea off the ice shelves, and therefore were not affected by the along-shelf dynamics assessed in our study. Yet, we found the same result regarding the importance of CAOs along the ice shelves and with summer observations. Jones et al. (2016) evaluated the performance of four reanalyses products in the Amundsen Sea, and showed that ERA-Interim has a cold bias in the air temperature, which is greater near the ice shelves and weaker far from the coast. As a consequence, they hypothesize that the heat loss would be overestimated. This hypothesis has been verified in our work with the bias found (Table 4) in ERA5 hybrid fluxes (computed from the atmospheric and sea surface ERA5 variables). However, the ERA5 nearest-neighbour THF underestimates the heat loss: this result indicates the importance of careful choice regarding the method used to retrieve estimate of turbulent flux in the Amundsen Sea from ERA5.

### 4.3 Implications

The Amundsen Sea is a dynamic shelf sea with horizontal processes such as coastal currents. Therefore the 1D PWP model does not aim to reproduce the observed temperature changes. However, we use the PWP model to give an insight on the potential effects of an air–sea THF misrepresentation on the seasonal evolution of SST in the Amundsen Sea. The stationary heat flux simulation and the hybrid simulation show similar results. This indicates that the large scale nature of the CAOs, which drive the strongest fluxes, negate any bias that might have been induced by simulating a 1D model using flux data from a moving ship. In the nearest neighbour simulation, which includes the strongly biased THF, there is a non-negligible impact with the





overestimation of SST (+ 0.76 °C) and HC ($8.2 \times 10^7$ J) at the end of the 35-days simulation. Yu et al. (2023) have shown that the climatological January SST ranges between -0.8 °C and -0.2 °C in the Amundsen Sea Polynya (it has an internal spatial variability), with an interannual standard deviation between 0.35 °C and 0.45 °C. The SST from the hybrid, stationary and observation simulations are therefore in the expected range unlike the nearest neighbour simulation (Fig. 9c). This could have a major impact regarding sea–ice formation or primary production studies as the sea–ice concentration variability is influenced by the SST in summer (Kumar et al., 2021), and the Chla concentration correlated to the SST in the Amundsen Sea (Garcia et al., 2021).

## 4.4 Outlook

A number of potential avenues of work became clear during the course of this study. (i) As mentioned earlier, it would be interesting to have THF observations when the ASL is further east, to better understand the dependence of flux variability on the ASL. This corresponds to the austral winter or autumn (Hosking et al., 2016): seasons where we currently lack observations. (ii) The air–sea heat flux is the sum of the turbulent and the radiative components. We hypothesize that the bias in ERA5 THF induced by the land–sea boundary would affect the radiative fluxes as well, with a misrepresentation of the albedo. (iii) Finally, we could expect numerical models with higher spatial resolution than ERA5 to better capture the THF magnitude and variability along the ice shelves. We could therefore compare the observations to some high resolution regional climate models (e.g. MetUM, RACMO) to evaluate their accuracy in a coastal shelf sea such as the Amundsen Sea.

*Code and data availability.* The meteorology and underway data of the research vessel, the Seaglider data and the CTD file are published at Zenodo (Queste et al., 2024). ERA5 data are available at the Copernicus Climate Change Service (C3S) Climate Data Store (CDS), doi: 10.24381/cds.adbb2d47, Hersbach et al. (2020). The AMSR2 sea–ice concentration data are made available at https://data.seaice.uni-bremen. de/amsr2/, Spreen et al. (2008). AirSeaFluxCode software is available at https://github.com/NOCSurfaceProcesses/AirSeaFluxCode/, doi: 10.3389/fmars.2022.1049168, Biri et al. (2023). RTopo-2 is available at https://doi.pangaea.de/10.1594/PANGAEA.856844, Schaffer et al. (2016). The MEaSUREs BedMachine Antarctica (V3) data set is accessible from the NASA National Snow and Ice Data Center Distributed Active Archive Center (NSIDC DAAC), doi: 10.5067/FPSU0V1MWUB6, Morlighem et al. (2020).

## Appendix A: Data processing: investigating the wind distortion effects

The anemometers on a ship can be positioned in a place where they experience airflow distortion from the superstructure of the research vessel (Yelland et al., 1998; Moat et al., 2005; Landwehr et al., 2020). The consequence is a bias in the wind speed values that depends on the location of the anemometers and the shape of the research vessel (Moat et al., 2005). We use ERA5 to analyse the residuals (i.e. research vessel minus ERA5 wind speed) and to validate the in situ wind measurements (Fig. A1).



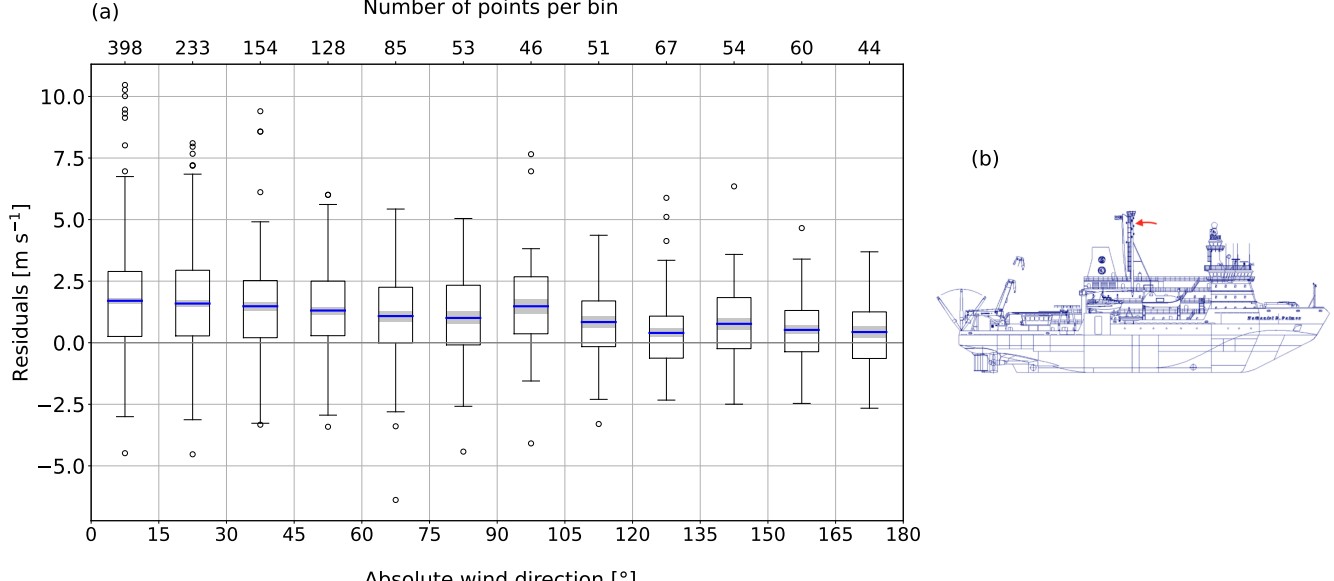

**Figure A1.** (a): residuals (10-meters wind speed[ship minus ERA5]) binned into 15 degrees relative absolute wind direction. "Relative" means relative to the ship, the convention is the following: a 0° relative wind direction means the wind is blowing bow-on. "Absolute" refers to the assumption that the flow is symmetrically distorted on the port and starboard side: e.g. the wind coming from (75, 90]° and from (-90, -75]° are distorted in the same way and therefore congregated in the same bin. The boxes extend from the lower to the upper quartile values, the whiskers represent the range of the data and the black empty circles are outliers. The blue lines are the mean per bin. The grey shaded areas indicate the standard error of the mean. (b) is RV *Nathaniel B. Palmer* (adapted from https://www.usap.gov/USAPgov/vesselScienceAndOperations/documents/NBP_Guide.pdf), the red arrow indicates the position of the anemometers.

We observe a decreasing residual with wind blowing from the stern, which could be explained by the openness of the superstructure over the back half of the ship (Fig. A1b). The residuals are positive for each bin, this is possibly an overestimate of the measured wind speed but also a bias from ERA5. We assume the bias is from ERA5 as it has been shown that it is biased low along the Antarctic coast (Caton Harrison et al., 2022). The mean residuals (blue lines, Fig. A1a) are low: the minimum is 0.40 m s$^{-1}$ for the bin (120, 135]° and the maximum is 1.70 m s$^{-1}$ for the bin (0, 15]° (winds blowing from the bow), we therefore decide to keep the data as the bias is not large.

## Appendix B: Reynolds decomposition

The Reynolds decomposition consists on the decomposition of a variable $X$ into the sum of its average ($\bar{X}$) and anomaly ($X'$). We applied this decomposition to the turbulent flux, and hereafter are the mathematical steps that led to the formulation of Equations 3 and 4.





$$\text{SHF} = \rho_{air} C_p (\overline{C_t} + C_t')(\overline{U} + U')(\overline{\Delta T} + \Delta T')$$

$$\text{LHF} = \rho_{air} L_v (\overline{C_q} + C_q')(\overline{U} + U')(\overline{\Delta q + \Delta q'})$$

We expand these equations and average them ($^-$). We use the following averaging rules:

$$\overline{X'} = 0$$

$$\overline{\overline{X}\overline{Y}} = \bar{X}\bar{Y}$$

$$\overline{\overline{X}Y'} = \bar{X}\overline{Y'} = \bar{X} \times 0 = 0$$

We obtain:

$$\overline{\text{SHF}} = \rho_{air} C_p \left[ \bar{C}_t \bar{U} \bar{\Delta T} + \bar{C}_t \overline{U'\Delta T'} + \bar{U}\overline{C_t'\Delta T'} + \bar{\Delta T}\overline{U'\Delta C_t'} + \overline{\Delta T'C_t'U'} \right]$$

$$\overline{\text{LHF}} = \rho_{air} L_v \left[ \bar{C}_q \bar{U} \bar{\Delta q} + \bar{C}_q \overline{U'\Delta q'} + \bar{U}\overline{C_q'\Delta q'} + \bar{\Delta q}\overline{U'\Delta C_q'} + \overline{\Delta q'C_q'U'} \right]$$

We then subtract the averaged $\overline{\text{SHF}}$ (resp. $\overline{\text{LHF}}$) to the total SHF (LHF) to retrieve the anomalous SHF' (LHF') of Eq. 3 (Eq. 4).




**Appendix C: PWP model input**

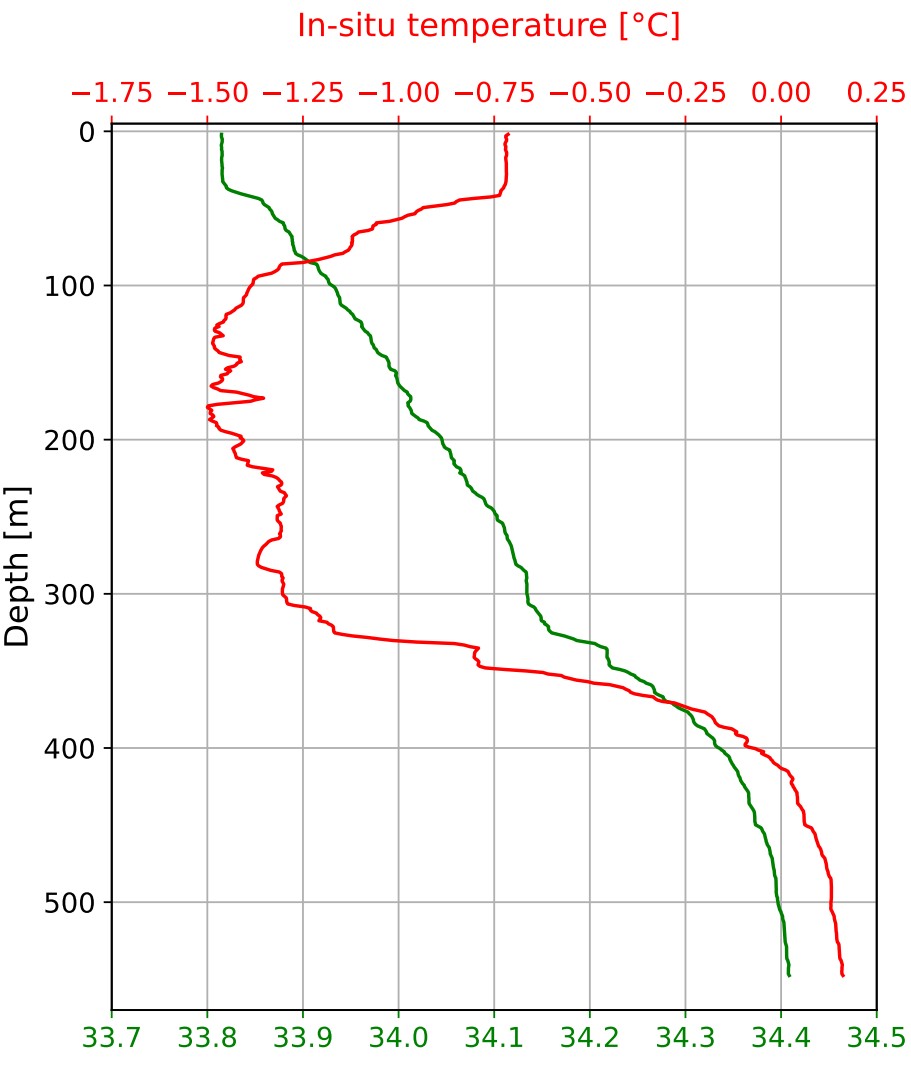

**Figure C1.** Oceanic forcing: initial temperature and salinity profiles for the PWP simulations from the CTD cast taken at 74.02° S, 113° W.





**Figure C2.** Atmospheric forcing: four runs were performed, only the latent and sensible heat flux differ. The shortwave, longwave and wind stress come from the research vessel observations. The precipitation comes from ERA5 stationary dataset because there was no observation of precipitation from the research vessel.

*Author contributions.* All coauthors defined the research problem and the conceptualization of the study. BJ carried out the data analysis and produced the figures and first draft under MdP and BYQ supervision as part of her MSc thesis. All coauthors discussed the analysis and contributed to the writing of the final manuscript.






*Competing interests.* The authors declare that they have no conflict of interest.

*Acknowledgements.* This work uses data collected on the TARSAN project, a component of the International Thwaites Glacier Collaboration (ITGC; https://thwaitesglacier.org/; NSF Grant 1929991; UKRI NERC Grant NE/S006419/1) and the ARTEMIS project (NSF Grant 1941483, and UKRI NERC Grant NE/W007045/1). BYQ thanks the support from the European Research Council (COMPASS, Grant No. 741120). MdP received funding from the European Union's Horizon 2020 research and innovation programme under grant agreement No. 821001 (SO-CHIC).



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
