# Peer review of "Turbulent heat flux dynamics along the Dotson and Getz ice-shelf fronts (Amundsen Sea, Antarctica)"

_EGUsphere, 2024_

## Author Comment (AC1)

**Reply on RC1**

The authors estimated the turbulent heat flux in the Amundsen Sea Polynya region during summer using in situ atmospheric data collected aboard the RV Nathaniel B. Palmer. They observed episodic heat loss events triggered by the outflow of cold, dry air from the Antarctic continent. A comparison with turbulent heat flux data from ERA5 revealed that ERA5, with its relatively coarse spatial resolution of 0.25 degrees, did not accurately reproduce the turbulent heat flux in the ocean along the ice shelf edge, leading to an underestimation. Heat flux estimates based on in situ observations in the Antarctic coastal areas, particularly in coastal polynya regions, are rare, making this study a valuable contribution to the polar science community. The data and analysis methods employed in this study appear to be reasonable. However, I have the following concerns and look forward to the authors' responses and revisions to the manuscript.

We thank referee #1 for careful reading and insightful comments. Throughout this document the initial comments from referee #1 are in black and our answers in blue.

This study emphasizes the importance of estimating turbulent flux due to its impact on heat loss and sea-ice production in coastal polynyas (*e.g.*, P. 1, L. 2–, P. 2, L. 37–). While this is undoubtedly true during the winter months, this study is based on summer observations. In winter, the dominant heat flux component is turbulent heat flux, whereas in summer, it is shortwave radiation, as shown in Fig. C2. This distinction should be clearly described in the manuscript. During summer, coastal polynyas act as "meltwater factories" due to solar heating of the upper ocean through open water with low albedo, contrasting with their role as "ice factories" in winter (Ohshima et al. ,1998 and Morales Maqueda et al., 2004). Therefore, I do not suggest removing the descriptions of coastal polynyas but rather believe they should be described with care. In recent years, the Antarctic sea-ice extent during summer has been unusually small (Purich and Doddridge, 2023). A prolonged open-ocean period in summer, resulting from anomalous sea-ice retreat, leads to increased solar heating and warming of the upper ocean, with this heat anomaly potentially influencing subsequent ice advance (Nihashi and Ohshima, 2001; Stammerjohn et al., 2012). The key factor here remains shortwave radiation, though heat loss to the atmosphere in autumn and winter is driven by turbulent heat flux. In the Amery Ice Shelf area, a reduction in summer sea-ice extent has been found to weaken the formation of Antarctic Bottom Water (Aoki et al., 2022). This is because anomalously small summer sea-ice extent leads to increased solar heating of the ocean, which accelerates the melting of the ice shelves and the supply of freshwater to the coastal polynya area, limiting the production of dense shelf water. Again, the primary heat flux component here is shortwave radiation, but turbulent flux also contributes to the

total heat flux. Given the significant changes occurring in the Antarctic sea ice, I believe that incorporating these perspectives could be valuable.

- Ohshima, K. I., K. Yoshida, H. Shimoda, M. Wakatsuchi, T. Endoh, and M. Fukuchi (1998), Relationship between the upper ocean and sea ice during the Antarctic melting season, J. Geophys. Res., 103, 7601–7615, doi:10.1029/97JC02806.
- Morales Maqueda, M. A., A. J. Willmott, and N. R. T. Biggs (2004), Polynya dynamics: A review of observations and modeling. Rev. Geophys., 42, RG1004, doi:10.1029/2002RG000116.
- Purich, A. and E. W. Doddridge (2023), Record low Antarctic sea ice coverage indicates a new sea ice state. Commun. Earth Environ. 4, 314, doi:10.1038/s43247-023-00961-9.
- Nihashi, S. and K. I. Ohshima (2001), Relationship between the sea ice cover in the retreat and advance seasons in the Antarctic Ocean, Geophys. Res. Lett., 28, 3677–3689, doi: 1029/2001GL012842.
- Stammerjohn, S, R. Massom, D. Rind, and D. Martinson (2012), Regions of rapid sea ice change: An inter-hemispheric seasonal comparison. Geophys. Res. Lett. 39, L06501, doi:10.1029/2012GL050874
- Aoki, S., T. Takahashi, K. Yamazaki, D. Hirano, K. Ono, K. Kusahara, T. Tamura, and G. D. Williams (2022), Warm surface waters increase Antarctic ice shelf melt and delay dense water formation. Commun. Earth Environ. 3, 142, doi:10.1038/s43247-022-00456-z

We thank referee #1 for providing perspectives and references to improve the contextualization of our study. We took into account their suggestion:

- In the abstract we state that sea–ice formation and melting occur in polynyas (and not only sea–ice formation):

  Initial sentence:

  *In coastal polynyas, where sea–ice formation occurs, it is crucial to have accurate estimates of heat fluxes in order to predict future rates of sea–ice formation.*

  Updated sentence (change in black):

  P. 1, L. 2–3 in the revised manuscript: *In coastal polynyas, where sea–ice formation* and melting occur, *it is crucial to have accurate estimates of heat fluxes in order to predict future sea–ice* dynamics.

- In the introduction we add background on the Amundsen Sea Polynya (opening mechanism, contribution to sea–ice formation in winter and melting in summer, size):

P. 2, L. 36–38: *Polynyas are defined based on their opening mechanism. The ASP is a wind-driven latent heat polynya that forms along the coastline. It has a mean open water area in the austral summer of 27,333 km$^2$ +/- 8749 km$^2$ and an average open duration of 131.9 +/- 17.5 days over 1997–2010 (Arrigo et al., 2012).*

P. 2, L. 39–43: *During winter, shelf water latent heat polynyas like the ASP usually inherit the "sea–ice factory" nickname (Morales Maqueda et al., 2004, Ohshima et al., 1998) because sea–ice is continually created and conveyed away by winds or currents. On the other hand, in summer the latent heat polynyas are "ice–melting factories" as the low albedo of open–water compared to surrounding sea–ice favours solar heating, resulting in melting sea–ice.*

P. 2–3, L. 59–62: *The turbulent heat flux is the main air–sea heat flux component during winter, whereas the radiative component dominates during summer (Morales Maqueda et al., 2004). Despite the importance of the radiative component in summer, key atmospheric conditions could set the scene for important episodic heat loss events.*

- In the discussion we add a paragraph on the implications of our work for sea–ice using the literature suggested by referee #1 (P. 26-27, L. 391–409).

Temperature and wind speed are crucial parameters for determining turbulent heat flux. A comparison of in situ observed wind speed with ERA5 data is shown in Fig. A1. How about including a similar comparison for temperature? As a reader, I believe such a comparison would provide valuable insights.

Fig. A1 main objective is to investigate the wind distortion imposed by the superstructure of the research vessel on wind speed values using ERA5 as a reference (evaluate if the residuals are decreasing/increasing with wind direction). In the initial manuscript, we didn't expect a temperature bias related to the wind direction and therefore didn't perform a similar comparison. When we perform the same analysis, comparing the ERA5 2-meter air temperature with the 2-m height adjusted air temperature from the research vessel show that there is indeed no bias in the 2-meters air temperature dependent on the wind direction (see Figure below).

[Figure]

However, as highlighted in the comment, Fig. A1 also shows a comparison of in situ observed wind speed with ERA5 (even if it wasn't the main point of this figure). Such comparison was not included in our study for air temperature. We do agree that a comparison between ERA5 and in situ observations would provide valuable insights. Such comparison has already been carried out in the Amundsen Sea by Jones et al., 2016 between in situ observations and ERA-Interim. We find biases with the same order of magnitude. We add the Figure below in the Appendix (Fig. D1) and discuss it in the subsection *4.2 Assessing ECMWF turbulent heat flux in the Amundsen Sea* (P. 26, L. 373–374): (the black sentence below is added, text in blue is copied for context)

*Jones et al. (2016) evaluated the performance of four reanalyses products in the Amundsen Sea, and showed that ERA-Interim has a cold bias in the air temperature and a dry bias in the specific humidity, which is greater near the ice shelves and weaker far from the coast. As a consequence, they hypothesize that the heat loss would be overestimated. This hypothesis has been verified in our work with the bias found (Table 4) in ERA5 hybrid fluxes (computed from the atmospheric and sea surface ERA5 variables).* This bias indeed arises from a cold and dry bias in ERA5 air and humidity (Figure D1b, c).

[Figure]

*Figure D1: Comparison of (y-axis) ERA5 and (x-axis) in situ observations for (a) wind speed at 10 meters, (b) specific humidity at 2 meters and (c) air temperature at 2 meters. The in situ observations were adjusted down to 10 and 2 meters thanks to AirSeaFluxCode which applies a logarithmic adjustment and stability functions to account for atmospheric stability. To determine specific humidity from ERA5, we convert the dewpoint temperature to specific humidity using the saturation vapour pressure function of Buck (2012). The scatters are coloured by the distance to the Amundsen Sea coastline in kilometers, that is defined as the closest coast point in the region lat = [73.66 ; 75.27]°S, lon = [108.10 ; 122.64]°W to the ship position.*

In Fig. A1, the wind speed from in situ observations is higher than that from ERA5. Could this discrepancy be due to the difference in observation heights, with the ship's measurements taken at 34.4 m (Table 1) and ERA5's at 10 m? A similar consideration applies to temperature: the ship's observations are taken at 19.2 m, while ERA5's are at 2 m. I suspect there may also be a bias in the temperature data. I believe the impact of these biases in wind speed and temperature on the turbulent heat flux estimates should be quantitatively discussed.

The sensor's heights (for in situ observations) or variable heights (for ERA5 data) of humidity, temperature and wind speed are input parameters of AirSeaFluxCode that need to be provided by the user. AirSeaFluxCode then computes the transfer coefficient $C_t(z_u, z_m)$ and $C_q(z_u, z_m)$ corresponding to the height $z_u$ of wind speed measurement and the height $z_m$ of humidity and temperature measurement using logarithmic corrections (we refer to Equations 1 and 3 of Biri et al., 2023). Any bias resulting from difference in observation heights is thus prevented.

The logarithm correction was very briefly explained on page 6, L. 95 of the initial manuscript, but only for the wind. We thank the reviewer for pointing out this lack of precision regarding the method used. We update Eq. (1) and (2) to make it clear that the transfer coefficients depend on the measured height of temperature, humidity and wind speed. The following sentence is also added to the manuscript (P. 7, L. 121–122):

*The AirSeaFluxCode applies a logarithmic correction in the transfer coefficient definitions ($C_t(z_m, z_u)$ and $C_q(z_m, z_u)$) to account for the height $z_u$ of wind speed and $z_m$ of air temperature and humidity measurements (Biri et al., 2023)*

AirSeaFluxCode allows the output of humidity, air temperature and wind speed at a reference height chosen by the user. For the example, we show below scatter plots of before (x-axis) and after (y-axis) the height correction from the sensors height (indicated in Table 1 of the manuscript) to 0.1 meters for (a) wind speed (b) specific humidity and (c) air temperature of the in situ data.

[Figure]

Regarding the estimation of turbulent flux (Eqs. 1 and 2), the influence of atmospheric stability on the heat transfer coefficient should be mentioned in this manuscript, even though it is discussed in the cited paper.

*The following sentence is added to the manuscript (P. 7, L. 122–123): "Atmospheric stability is accounted for in the definition of the transfer coefficients $C_t$ and $C_q$ through stability functions".*

Furthermore, since this study primarily focuses on the open ocean area of the summer coastal polynya region, I believe the influence is minimal. However, turbulent flux is also estimated in the sea-ice area (Fig. 3). In regions where sea ice and open water coexist, the estimation of turbulent flux is complicated by the significant thermal contrast between the sea ice, which acts as an insulator, and the open water. Additionally, considering atmospheric stability in such areas is challenging. How was the insulating effect of sea ice accounted for in the estimation of turbulent flux in this study?

*We thank the reviewer for raising this oversight. We didn't account for the insulating effect of sea ice in the initial manuscript due to the low importance of flux variability in sea ice for our study. To account for the reduced flux imposed by sea ice cover, we modify the initial flux computation by scaling the turbulent fluxes by the fraction of sea–ice concentration A when A > 15 % (Eqs. (3), (4) and (5)). For instance, the sensible heat flux would be modified in the presence of sea ice using (1-A)\*SHF,*

where A is the fraction of sea ice coverage and SHF is the sensible heat flux. We add the following sentences (P. 7, L. 131–134) to clarify this in the manuscript.

> *To account for the insulating effect of sea–ice we scale the turbulent fluxes by the sea–ice concentration (SIC) when SIC ≥ 15 % (Eq. (3), (4) and (5)). We acknowledge that this is a simplified method to account for sea–ice effect on turbulent fluxes, but accept this considering the little time spent by the RV in sea–ice covered area (3 days out of 57) and low importance of the flux variability in sea-ice for the results of this study.*

Minor comment:

1. 1, L. 3–: "The Amundsen Sea Polynya is the fourth largest coastal polynya ..." Is this referring to the size of the polynya or the sea-ice production?

This is referring to the size of the polynya and was obtained from Mu et al. (2014). However upon closer inspection we find their paper reference Arrigo et al., 2012 to provide the Amundsen Sea Polynya area but did not reference any sources that ranked Antarctic polynyas by size. For that reason, we remove this from our manuscript.

1. 1, L. 5: "NBP22/02" This expression makes sense to readers familiar with the observations by RV Nathaniel B. Palmer but is confusing to those who are not. It would be better to be more specific. Additionally, the description of the ship observation data begins on P. 2, L. 54, but the specific ship name does not appear until P. 3, L. 63. The ship name should be described earlier.

The acronym NBP22/02 was removed from the abstract and the specific ship name now appears at the beginning of the observations data section.

1. 6, L. 98: "... a heat loss (gain) for the ocean surface." would be appropriate.

It has been modified.

1. 7, L. 118: "... fresh water flux ..." During summer, the freshwater supply from melting sea ice is significant. Does this study consider that, or only precipitation?

The freshwater flux takes into account the evaporation and precipitation (P-E). The precipitation is an input of the PWP model whereas the evaporation is computed by the PWP model using the latent heat flux. We consider that sea–ice (which was already anomalously low in 2022) had already melted by the time the observations began and their impacts on sub-daily freshwater change is negligible over the time span of the research vessel stay in the Amundsen Sea. We add the following sentences on P. 8-9, L. 160–163:

*The input freshwater flux only contains precipitation and evaporation, freshwater input from melting sea–ice was not considered in this study. We consider this reasonable as 2022 was a record-low sea–ice year (Turner et al., 2022; Yadav et al., 2022), and most of the sea–ice melt had already occurred in the polynya region (https://data.seaice.uni-bremen.de/databrowser/).*

1. 8, Fig. 2c: It would be helpful if you could show the freezing point. Furthermore, since temperatures below 0°C are important, I would appreciate it if you could display them in a taller figure.

Fig. 2 has been updated.

References:

Arrigo, K.R., Lowry, K.E. and van Dijken, G.L., 2012. Annual changes in sea ice and phytoplankton in polynyas of the Amundsen Sea, Antarctica. *Deep Sea Research Part II: Topical Studies in Oceanography, 71,* pp.5-15.

Biri, S., Cornes, R.C., Berry, D.I., Kent, E.C. and Yelland, M.J., 2023. AirSeaFluxCode: Open-source software for calculating turbulent air-sea fluxes from meteorological parameters. *Frontiers in Marine Science, 9,* p.1049168.

Buck, A.L., 1996. Model Cr-1a hygrometer with autofill operating manual. *Buck Research Instruments LLC: Aurora, CO, USA*.

Jones, R.W., Renfrew, I.A., Orr, A., Webber, B.G.M., Holland, D.M. and Lazzara, M.A., 2016. Evaluation of four global reanalysis products using in situ observations in the Amundsen Sea Embayment, Antarctica. *Journal of Geophysical Research: Atmospheres*, *121*(11), pp.6240-6257.

Mu, L., Stammerjohn, S.E., Lowry, K.E. and Yager, P.L., 2014. Spatial variability of surface p $CO_2$ and air-sea $CO_2$ flux in the Amundsen Sea Polynya, Antarctica. *Elementa*, *3*, p.000036.

Turner, J., Holmes, C., Caton Harrison, T., Phillips, T., Jena, B., Reeves-Francois, T., Fogt, R., Thomas, E.R. and Bajish, C.C., 2022. Record low Antarctic sea ice cover in February 2022. *Geophysical Research Letters*, *49*(12), p.e2022GL098904.

Yadav, J., Kumar, A., Srivastava, A. and Mohan, R., 2022. Sea ice variability and trends in the Indian Ocean sector of Antarctica: Interaction with ENSO and SAM. *Environmental Research*, *212*, p.113481.

---

## Author Comment (AC2)

**Reply on RC2**

**Summary**

The authors make use of in situ observations taken from a research vessel located in the Amundsen Sea Polynya, Antarctica, to investigate ocean-atmosphere turbulent heat fluxes at this location. The authors examine the performance of the ERA5 reanalysis turbulent flux product in this region by comparing with the fluxes derived from observations. A 1D model is then used to evaluate the impact of differing turbulent heat fluxes on the sea surface temperature and heat content of the ocean mixed layer.

The paper is generally well-written, clear, and easy to read. It is interesting to see measurements of turbulent heat fluxes derived from in situ observations at the Amundsen Sea Polynya. Comparing these to the ERA5 reanalysis product provides a useful validation, and showing the impact of turbulent heat fluxes on SST and ocean heat content is also of value.

However, the paper is missing some detail on background, methods and implications of these results, which would make it more informative.

Questions asked below are intended as pointers to aid improvement of the paper, rather than solely as questions to answer in response to this review.

We thank referee #2 for their comments and suggestions for improvements. Throughout this document the initial comments from referee #2 are in black and our answers in blue.

**Major comments**

*Background*

The introduction should make clear what work in the paper is novel, and how it fits in with the existing literature. Literature should be referenced in the introduction to provide background and to put this research into context. What does this work add to the existing literature?

More general background on coastal polynyas should be included, for example could use Morales Maqueda et al. (2004) as a starting point: Morales Maqueda, M. A., A. J. Willmott, and N. R. T. Biggs (2004), Polynya dynamics: A review of observations and modeling, Rev. Geophys., 42, RG1004, doi:10.1029/2002RG000116.

More information on previous in situ observations at the ASP should be included as well as satellite observations and available reanalyses.

More background information on the ASP itself would be helpful. What is the polynya size? Frequency of occurrence? Does it change on the timescale of the field campaign?

In this paper, we add a new understanding of turbulent heat fluxes in the ASP, providing the first observational evidence of flux magnitude and variability across the summer period, and its impact on the seasonal evolution of SST. We have adjusted the final part of the introduction in L. 64–68 to make this clearer:

> "*We perform the first study of turbulent heat flux (THF) in the Amundsen Sea based on austral summer in situ observations and we (i) identify the temporal and spatial variability of the 2022 THF from shipboard observations, (ii) assess ERA5 accuracy at representing these fluxes, and (iii) investigate the relative importance of THF on the summer evolution of SST. Our findings provide evidence of the synoptic-scale air-sea interactions in the ASP and their impacts on seasonal evolution of SST.*".

More background regarding the polynya has been added to the manuscript:

- In the introduction (P. 2, L. 36–44) we added general background regarding coastal polynyas and specific background regarding the Amundsen Sea Polynya itself
- In the methods (P. 6, L. 104–107) some details have been added regarding the sea–ice evolution throughout the field campaign

> *It should be noted that during the research cruise, the sea ice gradually melted, so that from February 2022 onwards, we can no longer really speak of a polynya, as only a tongue of ice attached to the Thwaites ice shelf remains visible in the satellite-derived product (https://data.seaice.uni-bremen.de/databrowser/). We therefore refer to the polynya region.*

Regarding the request of adding more information on previous observations (in situ, from satellite) and reanalysis, we assume referee #2 wants more information on turbulent fluxes in the ASP. There has been no turbulent heat flux studies using in situ observations in the ASP, this is clarified in the manuscript (see above).

*Methods*

**Questions on the transfer coefficients used to compute the fluxes from observations**

What are the sensible and latent heat transfer coefficients ($C_t$, $C_q$) used for the COARE 3.5 algorithm? Are these valid for open ocean or sea ice? Do they change depending on the surface? How were they calculated? Is neutral stability being assumed? Is this a reasonable assumption?

Latent and sensible heat fluxes are computed using bulk parameterisations. $C_t$ and $C_q$ are the bulk transfer coefficients. They scale the exchange of heat (for $C_t$) and water vapour (for $C_q$) exchange at the air-sea interface. The coefficients depend on the height of measurement of wind speed and air humidity and temperature. Atmospheric stability is accounted for in the definition of $C_t$ and $C_q$. Neutral stability is therefore not assumed and this is a reasonable assumption. We refer to Edson et al., 2013 and Biri et al., 2023 for more details. We acknowledge that details regarding the transfer coefficients were missing in the initial manuscript and it has been modified in the revised manuscript: (P. 7, L. 121–123)

*The AirSeaFluxCode applies a logarithmic correction in the transfer coefficient definitions ($C_t(z_m, z_u)$ and $C_q(z_m, z_u)$) to account for the height $z_u$ of wind speed and $z_m$ of air temperature and humidity measurements (Biri et al., 2023). Atmospheric stability is accounted for in the definition of the transfer coefficients $C_t$ and $C_q$ through stability functions.*

The validity of the transfer coefficients over sea–ice is a concern also raised by referee #1 and that we address by scaling the flux by the sea–ice concentration (SIC), when SIC > 15%. While we acknowledge that this is a simplified method, we assume the impact is minimal as the RV spent only 3 days out of 57 in the sea–ice covered region. We added the following sentences in the manuscript: (P. 7, L. 131–134)

*To account for the insulating effect of sea–ice we scale the turbulent fluxes by the sea–ice concentration (SIC) when SIC ≥ 15 % (Eq. (3), (4) and (5)). We acknowledge that this is a simplified method to account for sea–ice effect on turbulent fluxes, but accept this considering the little time spent by the RV in sea–ice covered area (3 days out of 57) and low importance of the flux variability in sea-ice for the results of this study.*

**Questions on ERA5 fluxes**

How are ERA5 fluxes calculated? What heat transfer coefficients are used here? Is this different when the reanalysis thinks the surface is an ice shelf? Where does the ice shelf location dataset come from? What dataset does ERA5 use to define sea ice or open water? How is surface temperature determined? What validation has previously been performed on this product? What is the spatial and temporal resolution of this product? A spatial map of the data in the polynya region would be helpful.

ERA5 information regarding fluxes computation can be found in the documentation CY41R2 of the ECMWF Integrated Forecast System (IFS) Part IV - Physical processes. ERA5 fluxes are computed using bulk formulation as well. Just as the COARE 3.5 algorithm, the ECMWF algorithm uses transfer coefficients relying on stability functions and logarithmic adjustment for height of measurement. Even though the stability functions slightly differ between COARE 3.5 and ECMWF, the differences in the turbulent heat flux output is negligible compared to the differences between the observations and ERA5. Transfer coefficients rely on the Monin–Obukhov theory and consequently are functions of the roughness length. ECMWF turbulent fluxes are computed using the skin temperature. The roughness length is defined differently over land, sea or sea–ice in ECMWF. Antarctic RAMP2 (Radarsat Antarctic Mapping Project DEM Version 2, http://nsidc.org/data/docs/daac/nsidc0082 ramp dem v2.gd.html) is used by ECMWF to code the land fraction in Antarctica. A grid cell with a land fraction below or equal to 0.5 is a cell that is either open water or sea–ice. Sea–ice dataset comes from OSI SAF and surface temperature from OSTIA (Operational Sea-surface Temperature and sea Ice Analysis) product (Donlon et al., 2012).

We add a second panel on Fig. 7 showing the sea surface temperature from ERA5.

**Co-location method**

The co-location method to match up THF observations and the ERA5 reanalysis would be improved by using interpolation (e.g. nearest neighbour interpolation) of the reanalysis to the observation location, rather than a simple selection of the nearest neighbouring point. It would also be usual to perform a quality check on the data beforehand, which should remove points classified as the ice shelf in the reanalysis dataset.

To our knowledge, the nearest neighbour interpolation will include the information about the nearest ERA5 grid point to the THF observations, which would render the co-located ERA5 value non-comparable (due to it containing data on the ice shelf). We agree with the reviewer that one should perform a quality control check on the data beforehand and remove points classified as the ice shelf. This is one of the findings of the manuscript that we hope is emphasised, particularly when studying model data so close to ice shelves.

How was time co-location of the matchups of the THF observations with the ERA5 reanalysis performed? What is the temporal resolution of the ERA5 product?

The THF observations were averaged to hourly intervals to match the ERA5 resolution. We include this in (P. 5, L. 94–96) subsection *2.1.2 Reanalysis dataset: ERA5* :

> *It has an hourly temporal resolution and a regular 0.25° lat-lon grid. To co-locate its variables to the ship data, we use the nearest neighbour grid cell and the corresponding hour (as the ship data have been hourly averaged).*

Even if using the existing co-location method, why not simply choose the nearest ERA5 THF that is an ocean point, rather than recalculating the THF using ocean temperature and the COARE 3.5 algorithm?

This has been investigated but not presented in our manuscript. We didn't select the nearest ERA5 THF that is an ocean point as this method induces a bias due to the high spatial gradients in air temperature (Figure below).

[Figure]

Indeed, we show in our study that there is a strong spatial gradient in air temperature (Figure 5d), with colder air temperature near the ice shelves. We show below a comparison of the observed and ERA5 air temperature using (A)) the nearest neighbour grid cell that is an ocean point, and (B)) the nearest neighbour grid cell. We see that using the nearest neighbour grid cell on sea leads to a large overestimation of the air temperature (raw A)).

[Figure]

We thank the reviewer for pointing out the possibility to use the existing co-location method on ERA5 THF. We add a sentence in the manuscript to explain why we didn't select this method, P. 18, L. 245–246:

*We considered using the nearest ERA5 THF that is an ocean point, but this method was not chosen as this would induce a bias in the THF magnitude caused by an overestimation of air temperature.*

**Air and saturated humidity computation**

Should include the method used to calculate the air humidity and saturated humidity for the latent heat flux from the relative humidity recorded by the measurements (as given by Table 1).

AirSeaFluxCode requires humidity input in one of three formats: relative humidity [%], specific humidity [kg/kg] or dewpoint temperature [K] along with the specification of the format 'rh', 'q' or 'Td'. AirSeaFluxCode then converts the measured humidity to saturated humidity by using the saturation vapour pressure function from Buck (2012). We add the following sentence to the manuscript, P. 7, L. 123–124.:

*The measured relative humidity is converted to saturated humidity by using saturation vapour pressure function from Buck (2012).*

**Ocean heat content computation**

The method of calculation of the ocean heat content is also missing.

Indeed the equation is missing, we thank referee #2 for spotting this oversight. This is added to the manuscript (P. 9, L. 164–171):

*We compute the ocean HC (Eq. (9)) across the 40 meters upper layer because all ffour simulations converge below 40 meters (Fig. 9g). $\rho_0$ the mean potential density in the first 40 metres computed from Absolute Salinity and Conservative Temperature; $c_p$ the mean specific heat capacity in the first 40 metres computed from Absolute Salinity, in situ temperature and sea pressure and $C_T$ the Conservative Temperature.*

$$HC = \rho_0 \, c_p \int_{z=0}^{z=40} C_T \, dz$$

**Sea ice observations**

Were there no ship-board observations of sea ice concentration? What is the accuracy of the ASI product?

There were no recorded observations of sea ice concentration on the cruise. The original ASI algorithm uses Advanced Microwave Scanning Radiometer-EOS (AMSR-E) swath data. This product was validated against observations (Spreen et al., 2008). From July 2012 onwards the ASI algorithm was adjusted to the successor instrument AMSR2. We add the following sentence to the manuscript, P. 6, L. 103–104:

*The ASI algorithm has been validated against observations and shows good performance (Spreen et al., 2008).*

**Conversion of sea surface temperature to skin temperature**

Equation 1 refers to skin temperature of the ocean. However, the SST observations here are measured at 5 m depth, which is not the same thing. How will this impact the sensible heat flux calculation? A good description of the different SST definitions is here, if needed: https://www.ghrsst.org/wp-content/uploads/2021/04/SSTDefinitionsDiscussion.pdf

We apologise for any confusion and lack of clarity in the manuscript. In AirSeaFluxCode and the COARE 3.5 algorithm, the user inputs the SST and SST type: skin SST or bulk SST (bulk SST corresponds to the sea surface foundation temperature in https://www.ghrsst.org/wp-content/uploads/2021/04/SSTDefinitionsDiscussion.pdf).

If the bulk SST is given as input, which is the case in our study, the algorithm corrects the temperature for the warm layer and the cool skin to output the skin temperature before computing the sensible heat flux. This correction is made following Fairall et al., 1996.

To clarify this point, the following sentence is added in the manuscript (P. 7, L. 124–125):

*Warm layer and cool skin corrections are applied to convert the measured SST (Table 1) to skin SST as required in Eq. (1). These corrections follow Fairall et al. (1996).*

**Uncertainties discussion**

CTD and TSG measurements should also be discussed in the observations section. How accurate are the observations? Are uncertainty estimates available? What is the uncertainty on the results? How does this magnitude compare to the differences seen between results from using different methods?

The shipboard measurements of temperature used in this study from the TSG and the CTD are cross-calibrated with each other (see Figure below). The differences in temperature between the TSG and CTD would impose uncertainties on the flux calculation, but are an order of magnitude smaller (mean difference = 0.05 °C, Figure below) than the observed variability (std = 0.39°C in the polynya region, Fig. 2 of the manuscript).

[Figure]

*Implications*

What are the wider implications of the results? E.g. for biological production in the ASP or melt of the nearby glaciers? Dense water formation? Although the paper mentions that some of this would be affected, it is not stated how.

The Amundsen Sea is not a site of Antarctic Bottom Water formation (Moorman et al., 2020). The nearby Thwaite glacier is melting because of the intrusion of modified Circumpolar Deep Water (Arneborg et al., 2012). These two points are not discussed in the revised manuscript. However some other wider implications have been added in the subsection *4.3 Implications*:
- (P. 26–27, L. 391–403): implications of this study on sea–ice formation and melting, polynya duration and subsequent primary production
- (P. 27, L. 406–409): Implications on ice shelf basal melting from surface heating

What is the impact on heat loss over the whole polynya from the error in ERA5 ice shelf location vs observations? Is it just a small surface area missing from the turbulent flux? Does the fact that the area closest to the ice shelf experiences the highest fluxes change things? Can this impact be quantified? What is the impact over a year? On longer timescales? It is stated in the paper that the impact on sea ice formation, ice shelf melt and primary production is not negligible, but this needs to be expanded on and quantified in some way if possible.

The impact of the error in ERA5 air-sea heat flux for the total ocean heat gain in summer and loss in winter is an interesting question and has clear relevance to our study. Particularly given the importance of turbulent heat flux variability associated with rapid atmospheric dynamics. However, with the available data, our study is only relevant at the local scale and we do not extrapolate over the whole polynya or across other seasons. As such, we would only be able to speculate on the misrepresentation of air-sea heat exchange across the larger spatial scales and at longer time scales. We believe that our work provides a first step towards this important knowledge gap and should be investigated further. We anticipate that this would have implications for the melting of ice shelves. For instance, Stewart et al., 2019 show near the Ross Sea polynya that the ice shelf basal melting from surface heating is more important that what is traditionally thought and is expected to increase in the future. The fact that the area closest to the ice shelf experiences the highest fluxes implies that modelling studies relying on ERA5 turbulent heat flux could underestimate ice shelf melting. The quantification of this impact is beyond the scope of study.

**Minor comments**

Cold air outbreaks are mentioned in the literature review section, but the first half of the paper refers to this phenomenon as heat loss events. Terminology for these events should be made consistent throughout the paper.

Cold air outbreaks and heat loss are different. Heat loss corresponds to a transfer of heat by the ocean into the atmosphere whereas a cold air outbreak is an intrusion of cold air over a warm ocean. A cold air outbreak can lead to heat loss, but not all heat loss is due to cold air outbreaks. We acknowledge that this may have come across as confusing and add a definition of cold air outbreaks to prevent any further confusion (P. 2, L. 48–49):

*CAOs are equatorward intrusion of cold air over the warmer ocean (Papritz et al., 2015).*

The validation of the ERA5 reanalysis product highlighted a mismatch between the ice shelf edge in the product and in reality. While it is useful to point this out, this is not unusual, due both to the resolution of the product, and the fact that ice shelf edges are not static over

time. It looks from Figure 7 like the ship is not located at any point designated 100% land. This implies that the issue is with the spatial resolution of the product, rather than the ice shelf mask used for the ERA5 product being out-of-date.

We thank the reviewer for picking up on this and agree that it needs to be clarified. To avoid any confusion we modify the manuscript:

- In the abstract

The heat loss is larger along the ice shelves, and it is also where the ERA5 turbulent heat flux exhibits the largest biases, underestimating the flux by up to 141 W m$^{-2}$ due to its coarse resolution .

- In the results (3.2.1 Turbulent heat flux bias at land–sea boundaries)

The low agreement between the two flux products is explained by  the coarse resolution of ERA5 at land–sea boundaries.

It would be useful to show the RMS difference between the products in Table 4, as this would aid in the interpretation of the differences between Fig. 8b,d and Fig. 6b,d (line 211). Note also that these difference might reduce if using interpolation in the co-location method (see above).

The RMS difference is added in Table 4.

The distributions of the different results shown on Figure 10 should be discussed further.

We are unsure of the particular discussion items that the reviewer has in mind regarding Figure 10. For the sake of brevity, we refrain from adding discussion material that we do not believe adds value.

The discussion in Section 4.1.1 should be worked in to the relevant parts of the paper rather than separated out, e.g. see comment about line 97 below.

We thank referee #2 for this suggestion that we take into account:

- Part of the discussion is moved to the subsection *The role of the broad atmospheric system.*
- Lines 291-293 are moved to the methods section as suggested (comment about the line 97 below)

Much of the discussion of relevant literature in Section 4.1.2 should instead form part of the introduction. This would provide background for the research and then context for the discussion of results in this section.

We move part of Section 4.1.2 to the introduction and add a sentence about the relationship between sea-ice and the Amundsen Sea Low. The following paragraph is now in the introduction (P. 2, L. 45–50):

*Air–sea heat fluxes must also be considered in the context of the broader atmospheric circulation. In the Southern Ocean near the Polar Front, at 54° S, 89° W, Ogle et al. (2018) show that the advection of cold and dry air triggers ocean heat loss. In the Amundsen Sea, Papritz et al., (2015) show from ERA-Interim data that the Amundsen Sea is a hotspot for cold air outbreaks (CAOs), which contribute to the turbulent loss. CAOs are equatorward intrusion of cold air over the warmer ocean (Papritz et al., 2015). The large-scale atmospheric system also impacts sea–ice: in 2022 the Amundsen Sea Low (ASL), a quasi stationary low pressure center, has enhanced sea–ice melting (Turner et al., 2022; Yadav et al., 2022).*

**Specific (minor) comments**

Line 4: Differentiate between in situ and satellite observations – ASP is only poorly observed through in situ observations.

Done.

Line 4: Models and reanalyses are not wholly unvalidated, clarify that what is meant here is against in situ observations only

Done.

Line 8: "along the ice shelves" should say "along the ocean in front of the ice shelves" or similar, for clarity, as it's not the ice shelves themselves losing heat

Done:  *"The heat loss is larger along the ice shelves"* became  *"The ocean heat loss is larger along the ice shelf front"* (P. 1, L. 8).

Line 12: underestimation of 28 W m$^{-2}$ perhaps? (double negative if -28 W m$^{-2}$). This is similar on lines 271-2 (heat loss of -230 W m$^{-2}$), but you may prefer to leave it as it is.

The minus sign has been removed from Line 12 and Line 272, thank you for the suggestion.

Line 31: Reference Figure 1 in this sentence, e.g. "The Amundsen Sea, West Antarctica (Figure 1)…"

Done.

Line 37: More description of coastal/wind-driven polynyas and "ice-factory" mechanism is needed here, including the continual off-shore transport of ice by winds.

Done, the following sentences have been added in the introduction:

   *(P. 2, L. 36–37): Polynyas are defined based on their opening mechanism. The ASP is a wind-driven latent heat polynya that forms along the coastline.*

   *(P. 2, L. 39–41): During winter, shelf water latent heat polynyas like the ASP usually inherit the "sea–ice factory" nickname (Morales Maqueda et al., 2004; Ohshima et al., 1998) because sea–ice is continually created and conveyed away by winds or currents.*

Line 39: Clarify here whether the observations in this study are the only in situ turbulent heat flux observations available for the ASP

We agree this should be clarified. This is added in the introduction (see our answer above to the comment about the background).

Line 54: Clarify that THF includes sensible and latent heat fluxes (it's mentioned later in section 2.2.1 but the question first arises here)

Line 54 is a subsection title (2.1.1 Observations: shipboard and glider data). We believe referee #2 meant this should be added in the introduction. The following sentence is modified (modification in black), P. 2, L. 59–60:

The air–sea heat flux has two components: the radiative flux (sum of the shortwave and longwave radiations) and the turbulent flux (sum of the sensible and latent fluxes).

Line 55: What is an "underway system"?

The "underway system" is replaced by "the thermosalinograph".

Line 56: Clarify that it is the "bulk" turbulent heat flux being calculated, rather than using an eddy covariance method using high-frequency observations.

Done.

Line 56: "Punta Arena" should be "Punta Arenas"

Done.

Line 59: What is the sampling frequency of the observations from which the hourly averages (would be better to state "means" here, rather than "averages") are made?

Done, this has been modified (modifications in black), P. 3, L. 76–77:

To determine the THF and be consistent with ERA5's temporal resolution, we compute hourly means of the variables in Table 1. The initial resolution was 1-minute.

Line 61: It should be mentioned that there will be a positional bias in these observations (in a different way to e.g. airborne data) as the ship presumably won't travel into regions of higher concentration sea ice. What is the maximum SIC for this dataset? (Note that polynyas can be covered in high concentration, but thin ice, with large heat fluxes still associated with them)

While the RV certainly avoided travelling into regions of higher sea ice concentration, this does not impact the results presented in this study. The maximum SIC for this dataset is 95 % on the 26 Feb. 2022. We add the following sentence: P. 3, L. 80–81

The RV Nathaniel B. Palmer presumably avoided regions of higher sea ice concentration on its transit to the Amundsen Sea.

Line 63: Define RV acronym

Done, see Line 71 of the revised manuscript.

Line 66: What depths did the glider cover? How close to the surface did it take measurements? Why 40 m for the HC calculations – does this provide good coverage of the mixed layer?

We added the details below: (P. 4, L. 87–88)

*It sampled to the seabed, with a maximum of 901 meters, surfacing between each dive.*

Details regarding the HC calculations have been added in the method section (see above, our response to the comment about the ocean heat content computation).

Figure 1: Suggest making the inset map a bit bigger. What is the grey in Figure 1(b)? CTD should be mentioned in the observation section too, and the acronym defined.

The inset map is modified to be bigger. Values of sea–ice concentration equal to 0 were masked, the grey is a background colour. This is modified in the new figure to keep values that equal 0. A sentence about the CTD is added in the section describing the observations: P. 3, L. 82–85.

*Several Conductivity Temperature and Depth (CTD) casts were taken during the research campaign. In the present study we use one, taken at 74.02°S, 113°W in front of Dotson Ice Shelf (Fig. 1a, blue square) to initialise the 1D PWP model (see the model description in Sec. 2.2.2).*

Line 80: "ASMR2" should be "AMSR2"

This is modified, thank you for noticing this typo.

Line 87: Suggest rewording "the evaporation" to "sea surface evaporation" or similar, for clarity

Done.

Line 95: Does the logarithmic wind speed adjustment assume neutral atmospheric stability?

No, this has been modified (see our answer to the questions on the transfer coefficients above).

Line 97: What impact on the overall results would neglecting ocean surface velocity have? Suggest move lines 291-293 here.

Neglecting rapid surface currents during periods of relatively low winds speeds can alter the bulk air-sea flux calculations by up to 10% in highly turbulent ocean regions (Dawe and Thompson et al. 2006). However, the effect is negligible is low current speeds and high winds. As such, we did not take into account the impact of ocean surface velocity as we do

not have velocity measurements available. We consider this assumption reasonable due to the ocean velocity in this region being typically two orders of magnitude lower than the surface wind speeds (Carvajal et al. 2013). Lines 291–293 were moved as suggested.

Line 109: Suggest changing "/=" notation as this implies "does not equal"

Done.

Line 114: Suggest changing section title to include "the heat content of the mixed layer"

The integration of the upper ocean temperature to determine the heat content was performed across the top 40 m. As the mixed layer is occasionally shallower than 40 m, we do not believe changing the title would accurately represent the findings of this section.

Line 116: Add "produced" before "using observations and ERA5" for clarity.

Done.

Line 119: "blue rectangle" is described as a square on figure caption, should be consistent

Thank you for noticing this, we change "rectangle" to "square".

Figure 2: Were there any sea ice covered points which were also along the ice shelf front? The colour coding method doesn't allow for this, so it should be clarified if this was the case for any points. Clarify also that the "main sea surface and atmospheric variables" are from the ship observations. Suggest choosing a different colour for the blue area in (f) as it looks similar to the grey, and to the Southern Ocean classification too.

There were no sea ice covered points detected near the ice shelf front. This looks reasonable as the ASP is a wind-driven polynya so the sea ice is continually advected away from the ice shelf. We add the following sentence in the caption:

*Note that no sea ice was found near the ice shelf front.*

We change the colour in (f) for clarity.

Line 131: How are the uncertainties calculated? These are very large compared to the magnitude – this should be commented on.

The uncertainties are not computed, we provided the standard deviation of the measurements.

We change $-52 \pm 51\,W\,m^{-2}$ to $-52\,W\,m^{-2}$ ($STD = 51\,W\,m^{-2}$) in order to avoid further confusion (P. 11, L. 179).

Figure 3: caption "latter" is not needed here

"Latter" has been removed from the caption.

Line 135: What is the reason for the differing domination of LHF and SHF in different regions?

The SHF domination in the Amundsen Sea is unravelled in the rest of the manuscript (cold air outbreaks drive episodic sensible heat loss events). We do not explicitly state why the LHF dominates in the Southern Ocean region (Yu et al., 2012; Liu et al., 2011). However, this may be due to different atmospheric weather systems driving the flux variability in the open ocean (away from the ice shelf) such as extratropical cyclones (e.g. Ogle et al. 2018). We do not investigate this explicitly and so do not discuss it in the manuscript.

Figure 4: Suggest replace "on top of the black rectangles" with "outside of the black rectangles" as some are shown underneath. Suggest could show empty (-130, -90] bin for LHF so that the two figures line up.

Modification done for the caption. We update Figure 4 to show the empty (-130, -90] for LHF.

Line 148: "each time step" – suggest rewording this as it implies a model. Calculated for hourly data?

We replaced "each time step" by "each hourly data".

Table 3: "indice" should be "index"

We believe we should keep "indice" and not "index" for consistency throughout the manuscript (e.g., L. 109, 112, 147, 152, 196).

Line 166: Mention that it's the grey shaded area shown on Figure 3.

This has been added.

Line 169: add "visible on the map (Fig. 5d)"

We add Fig. 5d to the sentence.

Line 172-3: Should this be Fig. 3a,c not Fig. 3a,b? Fig. 3e might be better to refer to though.

We agree, thank you for noticing this referencing mistake. We change Fig. 3a,c to Fig. 3e.

Line 179: Suggest reword "comparing them to the COARE 3.5 fluxes from the observations" to "comparing them to the observed fluxes, calculated using the COARE 3.5 algorithm" for clarity.

We agree, this is done.

Figure 6 caption: Clarify that where SST is NaN it is classed as ice shelf. What happens to the SST if the surface is covered in sea ice?

We modified Figure 6 caption:

   *(in blue the data points where the ERA5 cell has a SST value, in yellow where SST is NaN: these points are then classed as ice shelf)*

If the surface is covered in sea ice we still have a SST value (for the observations and for ERA5), an example is shown below for ERA5 on the 1st of January 2022.

[Figure]

Line 184: What is the ERA5 product temporal frequency? Would it be expected to capture hourly events?

As stated in the methods, ERA5 has an hourly resolution, therefore we expect it to capture hourly events.

Table 4: Suggest swapping LHF and SHF columns as this is the order in which they are discussed in the text. Suggest changing column header to e.g. "LHF max diff to obs" for clarity. Which way round is the difference calculated, e.g. obs minus product? Define "ds".

We implement these suggestions. We provide the maximum absolute difference, as stated in the caption, therefore the way round of the computation does not matter. We replace "ds" by "dataset".

Line 210: The mean LHF difference to observations in Table 4 is actually higher for the hybrid version (though very small) – this should be addressed in the text. What is the uncertainty range of these results?

Thank you for noticing this point. We update the text:

Initial sentence:

*The hybrid heat fluxes were also on average closer to the observations-based fluxes but with a negative bias: the mean LHF and SHF for the hybrid-based fluxes were lower by 3 and 4 W m$^{-2}$ than fluxes from the observations (Table 4) because of colder air–temperature (not shown).*

*Updated sentence: P. 19, L. 257–260.*

*The hybrid SHF was on average closer to the observations-based fluxes (lower by 3 W m$^{-2}$, Table 4) but with a negative bias because of colder air–temperature (Fig. D1c). Regarding the mean LHF, the hybrid dataset did not bring a clear improvement: the mean hybrid SHF is lower by 4 W m$^{-2}$ than SHF from observations whereas the mean ERA5 SHF output was higher by 3 W m$^{-2}$.*

Figure 9: The colour scale is quite hard to see on the plots when comparing the three simulations, particularly the palest colour. Is it possible to use more contrasting colours?

We modify the colours to make them more contrasting.

Line 239: What is the wider implication of this? (See comments above).

Line 239 refers to the impact of a 2-days THF overestimation on the SST and HC. We add the following sentence: P. 22, L. 289–290

*The cumulative effect of such events is critical in setting the SST and HC differences between the simulations throughout the 35-days simulations (Fig. 9e, f).*

Lines 245, 258, 340: Should replace "seasonal" with "over a month" or similar wording, as 35 days is not a full season.

This would be modified.

Line 245-6: "not negligible" – this needs to be expanded upon. What impact would this have on these processes? Can it be quantified? (See comments above).

We add a paragraph in the discussion, subsection *4.3 Implications*, on the impact of underestimated heat fluxes for sea–ice formation and melting, ice shelf melting and primary production (see answers to comments above).

Line 253: Should replace "less important" with "smaller"

It has been updated.

Figure 10: SST depth is given as 6 m here (and in Table 5), whereas in Table 1 it is quoted as ~5 m. Which is correct?

6 meters depth is correct, we corrected Table 1.

Line 275: Is 12 W m$^{-2}$ within the range of uncertainty?

12 W m$^{-2}$ is not within the range of uncertainty. A commonly cited uncertainty requirement is 10 W m$^{-2}$ for the net heat flux over a season (Bradley and Fairall, 2007; Fairall et al. 1996). To meet this goal, an uncertainty of 5 W m$^{-2}$ is required for the bulk turbulent fluxes (Bradley and Fairall, 2007). Brodeau et al., 2017 compared ECMWF, COARE 3.5 and NCAR algorithms and show that the major sources of discrepancies and uncertainties between the 3 algorithms arise from transfer coefficient definitions and cool skin correction. In this respect

ECMWF and COARE 3.5 show similar results, unlike the NCAR algorithm, because they account for cool skin and warm layer effect and convective gustiness which reduce uncertainties.

Line 283: "any pattern of wind intensity" – this wording is a bit confusing, perhaps "any pattern of increasing wind intensity" or similar

Done.

Line 330: Add something along the lines of "as the air was cold and dry enough to enhance the air-sea temperature and humidity gradients" despite it being summertime.

We implement this suggestion.

Line 333: Is ERA5 likely to be the same as ERA-Interim in this respect?

Thank you for this question. We add a Figure in the Appendix (Fig. D1) to show that ERA5 is indeed the same as ERA-Interim. A sentence is also added in the discussion (in black below): P. 26, L. 373–374.

> *This hypothesis has been verified in our work with the bias found (Table 4) in ERA5 hybrid fluxes (computed from the atmospheric and sea surface ERA5 variables).* **This bias indeed arises from cold and dry biases in ERA5 air and humidity (Fig. D1).**

Line 349: Define Chla.

Chla has been changed to chlorophyll-a.

Figure A1: How are points outside of the range defined as outliers? (Meaning, why are they not included within the range?)

The boxplot spans from the first quartile (Q1) to the third quartile (Q3) with the whiskers extending from Q1 - 1.5(Q3-Q1) to Q3 + 1.5(Q3-Q1). In this respect, outliers are data points lying outside the interval [ Q1 - 1.5(Q3-Q1)  ; Q3 + 1.5(Q3-Q1) ].

We update the caption to define Q1 and Q3 (change in black):

> *The boxes extend from the lower (Q1)  to the upper (Q3) quartile values, the whiskers represent the range of the data and the black empty circles are outliers.*

And we add the following sentence (in the caption as well):

> *The outliers are defined as data points lying outside the interval [ Q1 - 1.5(Q3-Q1)  ; Q3 + 1.5(Q3-Q1) ].*

Source: **https://matplotlib.org/stable/api/_as_gen/matplotlib.pyplot.boxplot.html**

Line 384: Typo in LHF equation: I believe the overbar should not extend over the delta_q'

Indeed, thank you for noticing this typo. This is modified in the revised manuscript.

**References**

Arneborg, L., Wåhlin, A.K., Björk, G., Liljebladh, B. and Orsi, A.H., 2012. Persistent inflow of warm water onto the central Amundsen shelf. *Nature Geoscience*, *5*(12), pp.876-880.

Biri, S., Cornes, R.C., Berry, D.I., Kent, E.C. and Yelland, M.J., 2023. AirSeaFluxCode: Open-source software for calculating turbulent air-sea fluxes from meteorological parameters. *Frontiers in Marine Science*, *9*, p.1049168.

Bradley, E.F. and Fairall, C.W., 2007. A guide to making climate quality meteorological and flux measurements at sea.

Brodeau, L., Barnier, B., Gulev, S.K. and Woods, C., 2017. Climatologically significant effects of some approximations in the bulk parameterizations of turbulent air–sea fluxes. *Journal of Physical Oceanography*, *47*(1), pp.5-28.

Buck, A.L., 1996. Model Cr-1a hygrometer with autofill operating manual. *Buck Research Instruments LLC: Aurora, CO, USA*.

Carvajal, G.K., Wåhlin, A.K., Eriksson, L.E. and Ulander, L.M., 2013. Correlation between synthetic aperture radar surface winds and deep water velocity in the Amundsen Sea, Antarctica. *Remote Sensing*, *5*(8), pp.4088-4106.

Dawe, J.T. and Thompson, L., 2006. Effect of ocean surface currents on wind stress, heat flux, and wind power input to the ocean. *Geophysical Research Letters*, *33*(9).

Donlon, C.J., Martin, M., Stark, J., Roberts-Jones, J., Fiedler, E. and Wimmer, W., 2012. The operational sea surface temperature and sea ice analysis (OSTIA) system. *Remote Sensing of Environment*, *116*, pp.140-158.

Edson, J.B., Jampana, V., Weller, R.A., Bigorre, S.P., Plueddemann, A.J., Fairall, C.W., Miller, S.D., Mahrt, L., Vickers, D. and Hersbach, H., 2013. On the exchange of momentum over the open ocean. *Journal of Physical Oceanography*, *43*(8), pp.1589-1610.

Fairall, C.W., Bradley, E.F., Rogers, D.P., Edson, J.B. and Young, G.S., 1996. Bulk parameterization of air‐sea fluxes for tropical ocean‐global atmosphere coupled‐ocean atmosphere response experiment. *Journal of Geophysical Research: Oceans*, *101*(C2), pp.3747-3764.

Liu, J., Xiao, T. and Chen, L., 2011. Intercomparisons of air–sea heat fluxes over the Southern Ocean. *Journal of Climate*, *24*(4), pp.1198-1211.

Moorman, R., Morrison, A.K. and McC. Hogg, A., 2020. Thermal responses to Antarctic ice shelf melt in an eddy-rich global ocean–sea ice model. *Journal of Climate*, *33*(15), pp.6599-6620.

Morales Maqueda, M.A., Willmott, A.J. and Biggs, N.R.T., 2004. Polynya dynamics: A review of observations and modeling. *Reviews of Geophysics*, *42*(1).

Ogle, S.E., Tamsitt, V., Josey, S.A., Gille, S.T., Cerovečki, I., Talley, L.D. and Weller, R.A., 2018. Episodic Southern Ocean heat loss and its mixed layer impacts revealed by the farthest south multiyear surface flux mooring. *Geophysical Research Letters*, *45*(10), pp.5002-5010.

Ohshima, K.I., Yoshida, K., Shimoda, H., Wakatsuchi, M., Endoh, T. and Fakuchi, M., 1998. Relationship between the upper ocean and sea ice during the Antarctic melting season. *Journal of Geophysical Research: Oceans*, *103*(C4), pp.7601-7615.

Papritz, L., Pfahl, S., Sodemann, H. and Wernli, H., 2015. A climatology of cold air outbreaks and their impact on air–sea heat fluxes in the high-latitude South Pacific. *Journal of Climate*, *28*(1), pp.342-364.

Spreen, G., Kaleschke, L. and Heygster, G., 2008. Sea ice remote sensing using AMSR-E 89-GHz channels. *Journal of Geophysical Research: Oceans*, *113*(C2).

Stewart, C.L., Christoffersen, P., Nicholls, K.W., Williams, M.J. and Dowdeswell, J.A., 2019. Basal melting of Ross Ice Shelf from solar heat absorption in an ice-front polynya. *Nature Geoscience*, *12*(6), pp.435-440.

Turner, J., Holmes, C., Caton Harrison, T., Phillips, T., Jena, B., Reeves-Francois, T., Fogt, R., Thomas, E.R. and Bajish, C.C., 2022. Record low Antarctic sea ice cover in February 2022. *Geophysical Research Letters*, *49*(12), p.e2022GL098904.

Yadav, J., Kumar, A., Srivastava, A. and Mohan, R., 2022. Sea ice variability and trends in the Indian Ocean sector of Antarctica: Interaction with ENSO and SAM. *Environmental Research*, *212*, p.113481.

Yu, L., Zhang, Z., Zhou, M., Zhong, S., Lenschow, D.H., Li, B., Wang, X., Li, S., Wu, H. and Sun, B., 2012. Trends in latent and sensible heat fluxes over the southern ocean